# *Shigella* entry unveils a calcium/calpain-dependent mechanism for inhibiting sumoylation

Pierre Lapaquette[1,2†§], Sabrina Fritah[1,2†#], Nouara Lhocine[3,4], Alexandra Andrieux[1,2], Giulia Nigro[3,4], Joëlle Mounier[3,4], Philippe Sansonetti[3,4‡*], Anne Dejean[1,2‡*]

[1]Nuclear Organization and Oncogenesis Unit, Institut Pasteur, Paris, France; [2]INSERM, U993, Paris, France; [3]Unité de Pathogénie Microbienne Moléculaire, Institut Pasteur, Paris, France; [4]INSERM, U786, Paris, France

**\*For correspondence:**
philippe.sansonetti@pasteur.fr (PS);
anne.dejean@pasteur.fr (AD)

[†]These authors contributed equally to this work
[‡]These authors also contributed equally to this work

**Present address:** [§]PAM UMR A 02.102, Univ. Bourgogne Franche-Comté, AgroSup Dijon, Dijon, France; [#]NorLux Neuro-Oncology Laboratory, Department of Oncology, Luxembourg Institute of Health, Luxembourg, Luxembourg

**Competing interests:** The authors declare that no competing interests exist.

**Abstract** Disruption of the sumoylation/desumoylation equilibrium is associated with several disease states such as cancer and infections, however the mechanisms regulating the global SUMO balance remain poorly defined. Here, we show that infection by *Shigella flexneri*, the causative agent of human bacillary dysentery, switches off host sumoylation during epithelial cell infection in vitro and in vivo and that this effect is mainly mediated by a calcium/calpain-induced cleavage of the SUMO E1 enzyme SAE2, thus leading to sumoylation inhibition. Furthermore, we describe a mechanism by which *Shigella* promotes its own invasion by altering the sumoylation state of RhoGDIα, a master negative regulator of RhoGTPase activity and actin polymerization. Together, our data suggest that SUMO modification is essential to restrain pathogenic bacterial entry by limiting cytoskeletal rearrangement induced by bacterial effectors. Moreover, these findings identify calcium-activated calpains as powerful modulators of cellular sumoylation levels with potentially broad implications in several physiological and pathological situations.
DOI: https://doi.org/10.7554/eLife.27444.001

## Introduction

The post-translational modification by SUMO is an essential regulatory mechanism of protein function that is involved in most challenges faced by eukaryotic cells, ranging from cell communication to gene expression (*Cubeñas-Potts and Matunis, 2013*; *Flotho and Melchior, 2013*). Mammals express three functional SUMO proteins, SUMO1 and SUMO2/3, with the latter two being almost identical. The sumoylation machinery is composed of an E1 SUMO enzyme (the SAE1/SAE2 heterodimer), a unique E2 SUMO enzyme (UBC9), and E3 SUMO enzymes that enhance SUMO conjugation of specific targets. The steady-state levels of sumoylated substrates are critically regulated by the action of desumoylating enzymes, such as SENPs. Sumoylation is characterized by its highly dynamic and reversible nature, resulting in only a very small fraction of a given protein substrate being sumoylated in the cell at steady state level (*Nayak and Müller, 2014*). Whereas the vast majority of SUMO substrates identified so far in proteomic analysis are nuclear, a number of cytosolic and plasma membrane proteins can also be targeted by SUMO (*Hendriks and Vertegaal, 2016*). Cellular sumoylation levels relies on the fine equilibrium between conjugating and deconjugating activities and perturbation in this balance has been associated with disease processes, including cancer (*Seeler and Dejean, 2017*) and infections by pathogenic micro-organisms (*Mattoscio et al., 2013*; *Srikanth and Verma, 2017*). However, while information on the specific roles of the different SUMO E3 and SENP enzymes is accumulating, our knowledge of possible mechanisms regulating the global sumoylation/desumoylation equilibrium still remains highly fragmentary.

Post-translational modifications enable cells to dynamically react to stress or pathogenic agents by modifying quickly, locally and specifically the activity of key proteins. Highjack of protein post-translational modifications is emerging as a key strategy used by pathogens to survive and usurp the cellular machinery to their own benefit (*Ribet et al., 2010*). Whereas the interplay between SUMO and viral infection is relatively well characterized (*Everett et al., 2013*; *Mattoscio et al., 2013*), the molecular mechanisms by which sumoylation acts to limit bacterial infection are poorly characterized. *Listeria monocytogenes* facilitates its infection capacity by inducing both a proteasome-dependent and -independent decrease in the amount of SUMO conjugates in host cells. This effect has been attributed to the pore-forming toxin LLO that is sufficient to induce a proteasome-independent degradation of UBC9 (*Ribet et al., 2010*). Another study indicates that the enteropathogenic bacteria *Salmonella* Typhimurium affects sumoylation through upregulation of two microRNAs, miR30c and miR30e, that post-transcriptionally repress UBC9 (*Verma et al., 2015*). Conversely, it has been reported that two human pathogenic bacteria, *Anaplasma phagocytophilum* and *Ehrlichia chaffeensis*, promote SUMO modification of their own effectors to facilitate their intracellular survival (*Beyer et al., 2015*; *Dunphy et al., 2014*). Finally, we have shown recently that, at early stage of infection, *Shigella* can alter either positively or negatively the sumoylation status of a restricted set of transcriptional regulators involved in inflammation (*Fritah et al., 2014*).

In most cases, pathogenic micro-organisms manipulate sumoylation through interference with the SUMO enzymatic machinery. However the precise mechanisms by which pathogenic bacteria subvert the SUMO pathway enzymes and the nature of the relevant host SUMO substrates remain largely unknown. Here, we analyzed the sumoylation status of host proteins at late stage of *Shigella* infection that revealed a dramatic decrease in the global amount of SUMO conjugates in epithelial cells. Mechanistically, we demonstrate that this effect is, in large part, mediated by a calpain-dependent proteolytic degradation of the E1 SAE2 enzyme. We show that impaired sumoylation activity in host cells favors *Shigella* entry and identified RhoGDIα, a master negative regulator of the biological activities of small Rho GTPases, as an important SUMO substrate used by host cells to limit *Shigella* invasion. This work provides mechanistic insight into how sumoylation, by countering cytoskeletal rearrangement, impairs bacterial infection. In addition, it establishes calcium signaling as a novel and potent regulator of cellular sumoylation that may be relevant to transiently and/or locally alter sumoylation levels in several normal or disease states.

## Results

### *Shigella* infection inhibits sumoylation in epithelial cells in vitro and in the gut

To investigate the impact of *Shigella* infection on global sumoylation of host cell proteins, we followed the global pattern of proteins conjugated to SUMO1 and SUMO2/3 at timed intervals after infection (0 to 180 min). A gradual decrease in both SUMO1 and SUMO2/3 conjugates was observed in HeLa cells infected with the invasive *Shigella* strain M90T. An almost complete disappearance of the higher molecular weight SUMO species was visible after 180 min (*Figure 1A* and *Figure 1—figure supplement 1A*). These data are in agreement with a recent report showing impaired sumoylation in similar conditions (*Sidik et al., 2015*). A weak, yet consistent reduction in the level of SUMO1-modified proteins was readily visible 30 min post-infection as shown by a ~ 25% decrease in the total amount of SUMO1 conjugates and a concomitant ~20% accumulation of free SUMO1 (*Figure 1A*, *Figure 1—figure supplement 1B* and *Figure 1—figure supplement 1—source data 1*). The decrease in the total amount of SUMO2 conjugates after 30 min was, in contrast, more pronounced (*Figure 1—figure supplement 1A*), a finding in line with the higher dynamics of modification by SUMO2 compared to SUMO1 (*Saitoh and Hinchey, 2000*). Of note, no accumulation of free SUMO2/3 could be detected in our setting. SUMO1 and SUMO2/3 conjugates decreased in a multiplicity-of-infection-dependent manner (*Figure 1B* and *Figure 1—figure supplement 1C*). This loss in SUMO conjugates was not observed in cells infected with the non-invasive *mxiD* mutant that lacks expression of the type III secretion system (T3SS) (*Figure 1C* and *Figure 1—figure supplement 1D*). Moreover, the global reduction in protein sumoylation was impaired in cells treated with cytochalasin D, a drug that prevents actin polymerization and thereby completely blocks *Shigella* entry. This indicates that actin cytoskeleton rearrangements are required for *Shigella* to impair sumoylation

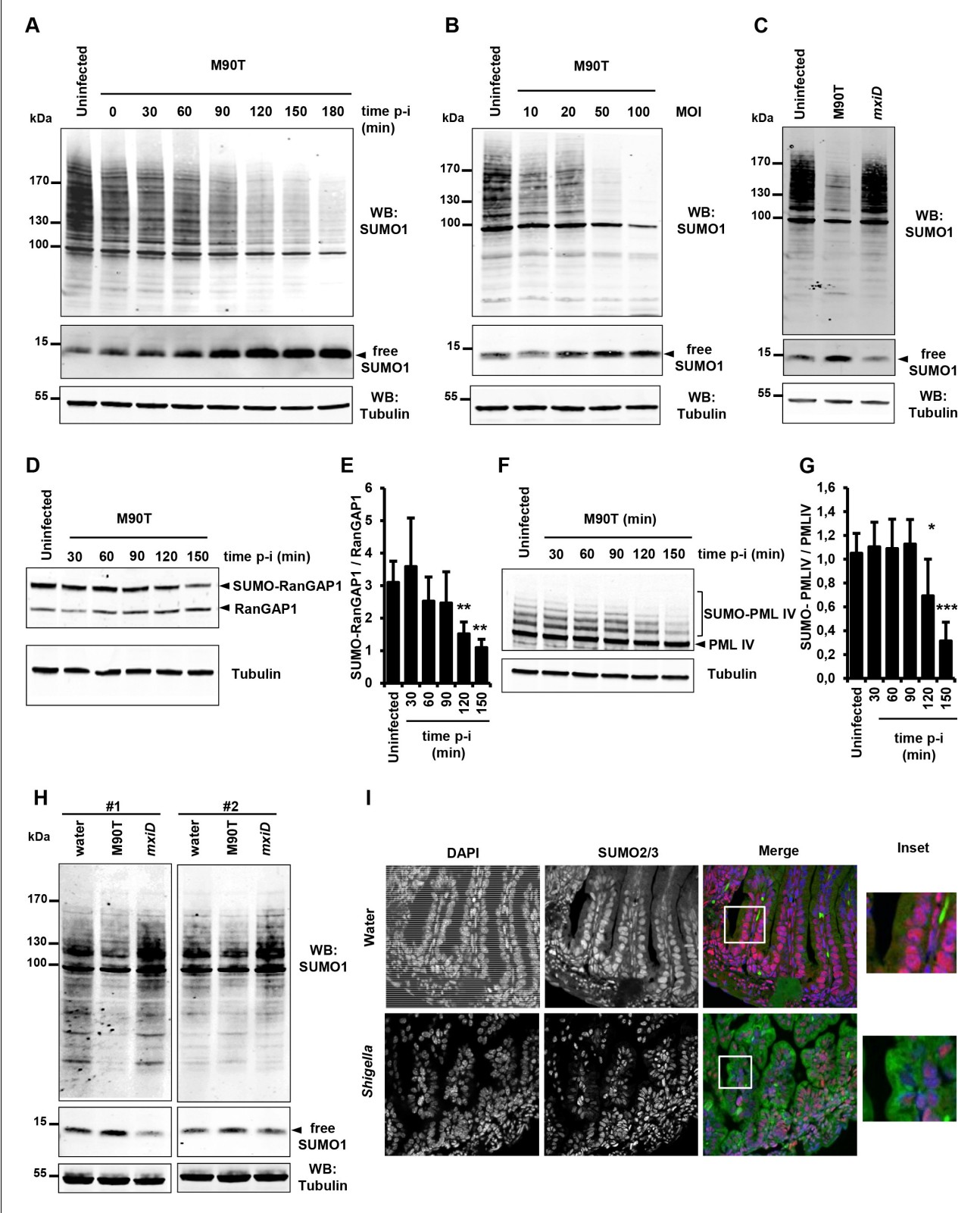

**Figure 1.** *Shigella* infection induces a massive loss in SUMO conjugates in vitro and in vivo. (**A**) SUMO1-conjugated protein patterns from uninfected HeLa cells or cells infected with the wild-type *Shigella* strain M90T for the indicated times. Immunoblot analysis was performed on whole-cell lysates using antibodies specific for SUMO1 isoform and tubulin. p-i: post-infection. (**B**) SUMO1-conjugated protein patterns from uninfected HeLa cells or cells infected with *Shigella* for 120 min at the indicated multiplicity of infection (MOI). (**C**) Global SUMO1 patterns of uninfected HeLa cells or cells infected

*Figure 1 continued on next page*

Figure 1 continued

with M90T or the *mxiD* non-invasive *Shigella* for 120 min. (D) Immunoblot analysis of RanGAP1 and SUMO-RanGAP1 levels in HeLa cells infected with M90T for the indicated times. (E) Quantification of the immunoblot signals are presented as SUMO-RanGAP1 signal relative to unmodified RanGAP1 signal (mean of four independent experiments ± s.d., *p<0.05, **p<0.01). (F) Immunoblot analysis of PML-IV and SUMO-PML-IV levels in HT1080 cells stably expressing GFP-PML-IV and infected with M90T for the indicated times. (G) Quantification of the immunoblot signals are presented as SUMO-PML-IV signal relative to unmodified GFP-PML-IV signal (mean of three independent experiments ± s.d., *p<0.05, ***p<0.001). (H) Global SUMO1 patterns of the whole intestine of 4-day-old newborn mice (#1 and #2), 180 min after inoculation of the invasive M90T strain or the non-invasive *mxiD* mutant. Physiological water was used as a control. Tubulin was used as a loading control. (I) Fluorescence microscopy analysis of the intestinal epithelium on paraffin sections after *Shigella* infection of newborn mice for 180 min. Physiological water was used as a control. SUMO2/3 appears in red, auto-fluorescence of the intestinal tissue in green and nuclei counterstained with DAPI in blue. White square, inset.

DOI: https://doi.org/10.7554/eLife.27444.002

The following source data and figure supplements are available for figure 1:

**Source data 1.** Quantification of the immunoblot signals relative to *Figure 1D–G*.

DOI: https://doi.org/10.7554/eLife.27444.006

**Figure supplement 1.** *Shigella* infection induces a massive loss in SUMO conjugates *in vitro* and *in vivo*.

DOI: https://doi.org/10.7554/eLife.27444.003

**Figure supplement 1—source data 1.** Source data file relative to *Figure 1—figure supplement 1B and E*.

DOI: https://doi.org/10.7554/eLife.27444.004

**Figure supplement 2.** Study of a panel of Shigella mutant strains for their ability to induce a loss in SUMO conjugates.

DOI: https://doi.org/10.7554/eLife.27444.005

(*Figure 1—figure supplement 1E–G* and and *Figure 1—figure supplement 1—source data 1*). We then validated the decrease in sumoylation on the two heavily sumoylated RanGAP1 and PML-IV substrates. In *Shigella*-infected HeLa cells, a reduction in sumoylated RanGAP1 was visible 90 min post-infection (*Figure 1D–E* and *Figure 1—source data 1*). Similar results were obtained in HT1080 cells stably expressing PML-IV where a significant decrease in SUMO-modified form of PML-IV occurred after 120 min infection (*Figure 1F–G* and *Figure 1—source data 1*). The hyposumoylation was more pronounced for PML-IV than for RanGAP1, in accordance with the high stability of the sumoylation state of RanGAP1.

To identify putative *Shigella* factors involved in the decrease in sumoylation, we tested a panel of mutant strains that affect bacterial virulence to various degrees for their ability to induce a loss in SUMO conjugates (*Figure 1—figure supplement 2A–C*). Among these mutants, the *mxiE* strain, that is defective for the expression and secretion of several important *Shigella* effectors (OspB, OspC1, OspD2, OspD3, OspE1, OspE2, OspF, OspG, VirA, IpaH1.4, IpaH4.5, IpaH7.8 and IpaH9.8) (*Bongrand et al., 2012*; *Kane et al., 2002*; *Mavris et al., 2002*), was still able to induce a decrease in sumoylated proteins, indicating that these different factors are not involved in sumoylation inhibition (*Figure 1—figure supplement 2A–B*). In a similar manner, mutation in the genes encoding the VirA, IpgD or OspG effectors did not affect the ability of *Shigella* to induce hyposumoylation (*Figure 1—figure supplement 2A–C*). In contrast, mutants for the expression of the translocator components IpaB and IpaC failed to induce a loss in SUMO conjugates, suggesting that the insertion of these bacterial proteins into the host plasma membrane is necessary to impair sumoylation (*Figure 1—figure supplement 2C*). We next analysed the effect of two insertion mutants of IpaC: *ipaC/pC351* and *ipaC/pC57* (*Mounier et al., 2009*). The *ipaC/pC351* mutant strain is unable to induce actin foci formation and to invade host cells, whereas *ipaC/pC57* is still able to form actin foci but is not able to efficiently invade host cells (*Mounier et al., 2009*). Only the *ipaC/pC57* mutant was able to induce hyposumoylation (*Figure 1—figure supplement 2C*), suggesting that, beyond pore formation, the stress induced by the bacteria at the plasma membrane actively contributes to the loss of sumoylation, while not requiring efficient invasion of host cells. Altogether, these results suggest that *Shigella* induces a loss in SUMO conjugates by triggering pore formation and subsequent plasma membrane stress.

To then see whether the findings obtained in vitro may transpose in vivo, we analyzed the SUMO patterns in the gut of newborn mice after 180 min infection with *Shigella*. A decrease in the amount of SUMO1-modified proteins together with an increase in free SUMO1 was clearly visible in mice infected with the invasive strain M90T compared to control or *mxiD*-infected animals (*Figure 1H*). Moreover, immunofluorescence staining of SUMO2/3 on paraffin-embedded sections of intestines

from M90T-infected newborn mice showed a marked decrease in SUMO2/3 staining in enterocyte nuclei compared to that observed in control newborn mice (**Figure 1I**). Altogether these results show that *Shigella* severely impairs sumoylation in vitro and that this effect occurs in vivo in the target organs of the pathogenic bacteria.

## Loss in SUMO conjugates is due to intracellular *Shigella*-induced activation of host calpain proteases

To gain insights into the mechanisms by which *Shigella* alters the sumoylation status of host cells, we probed a possible involvement of calpains. Indeed, *Shigella* infection of epithelial cells is known to rapidly activate calpains, a family of cysteine proteases known to cleave a wide range of substrates (*Bergounioux et al., 2012*). As expected, calpain activation was observed in response to *Shigella* infection as measured by the autolytic maturation process that converts the 30 kDa calpain regulatory subunit Capns1 into a truncated 18 kDa fragment (**Figure 2A**). In addition, we observed the degradation of the calpain endogenous inhibitor calpastatin (**Figure 2A**). Remarkably, pre-treatment of cells by the calpain inhibitor MDL28170 (*Mehdi et al., 1988*) entirely abrogated the loss of SUMO1- and SUMO2/3-conjugates upon *Shigella* infection (**Figure 2A**). In contrast, and in agreement with a previous report (*Bergounioux et al., 2012*), *Shigella*-induced calpastatin degradation was not suppressed by MDL28170 treatment, suggesting the involvement of another type of protease for its degradation. We then used two other systems for inhibiting calpain activity. Infection by *Shigella* of cells treated with siRNAs against Capns1 or infection of mouse embryonic fibroblasts (MEFs) knock-out for *Capns1* (*Capns1 KO*) similarly failed to trigger a loss in SUMO conjugates when compared to control cells (**Figure 2B and C**).

Calpain proteases are activated in response to an increase in intracellular calcium levels. It was reported that *Shigella* entry potently induces a local calcium response allowing cytoskeletal remodeling at early invasion stages. Shortly thereafter, it induces a global rise in calcium levels that enhances bacterial invasion and dissemination (*Bonnet and Tran Van Nhieu, 2016*). We thus assessed the direct effect of altering calcium levels on *Shigella*-mediated hyposumoylation. Treatment with the calcium-chelating agent BAPTA-AM, that blocks the release of calcium from intracellular stores, was sufficient to avoid calpain protease activation and subsequent loss in SUMO conjugates (**Figure 3A**). Conversely, treatment alone with calcium together with the calcium ionophore ionomycin, that increases the intracellular calcium levels, recapitulates the hyposumoylation seen upon *Shigella* infection. Such an effect was visible both at the global or the single substrate level, as exemplified by PML-IV (**Figure 3B–C**). Moreover, pre-treatment of the cells with the calpain inhibitor MDL28170 prevented the loss in SUMO conjugates induced by calcium and ionomycin (**Figure 3B–C**). Overall, these results demonstrate that increased calcium responses and subsequent calpain activation drives *Shigella*-induced loss of SUMO conjugates, indicating that calcium- and SUMO-dependent signaling are linked.

## SAE2 is a direct calpain substrate

As *Shigella* infection leads to a massive decrease in a large number of SUMO conjugates at late time post-infection, we hypothesized that calpain protease activity could target key proteins of the sumoylation machinery. We examined the levels of E1 and E2 SUMO enzymes in HeLa cells upon *Shigella* infection. Whereas the levels of SAE1 and UBC9 were not affected, a strong decrease in the level of SAE2 was observed (**Figure 4A**). The degradation of SAE2 triggered by *Shigella* infection was totally abrogated in cells treated with the calpain inhibitor MDL28170 (**Figure 4A**). In a similar manner, the loss of SAE2 was prevented in cells transfected with siRNAs against Capns1 or in *Capns1 KO* MEFs (**Figure 4B–C**). Thus the E1 enzyme SAE2 is proteolytically degraded by calpains upon *Shigella* infection, leading to the observed massive decrease in SUMO conjugates. Intriguingly, whereas the loss in SAE2 was barely visible before 2 hr post-infection (**Figure 4C**), the decrease in global sumoylation started as early as 30 min (**Figure 1A** and **Figure 1—figure supplement 1A,B**). The mechanisms responsible for the early reduction in SUMO conjugates triggered by *Shigella* infection remain to be elucidated (see Discussion). A direct involvement of increased SENP activities seems unlikely as the loss in SUMO-modified proteins is, like SAE2 loss, strictly dependent on calpains (**Figure 2A–C**), and no link has been so far established between calpain and SENP activities. In line with this, we failed to detect any significant desumoylase activities in total non-denatured

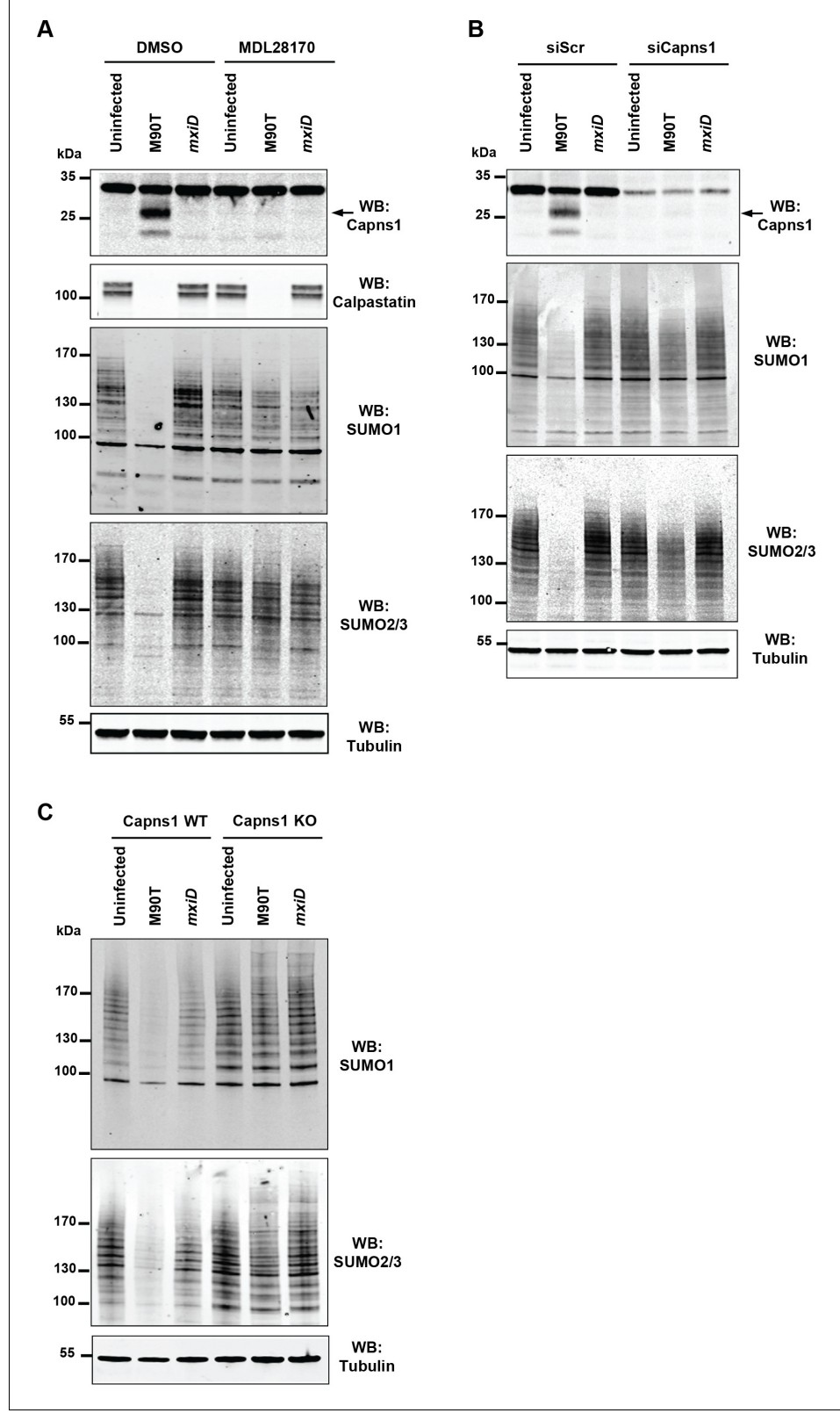

**Figure 2.** *Shigella* inhibits sumoylation by a calpain-dependent mechanism. (**A**) HeLa cells were pretreated by vehicle (DMSO) or 100 μM MDL28170 for 1 hr and then left uninfected or infected with the M90T strain or the *mxiD* mutant for 2 hr. Immunoblot analysis were performed using anti-Capns1 (Calpain small subunit 1), -calpastatin, -SUMO1, -SUMO2/3 and -tubulin antibodies. The 18 kDa truncated Capns1 fragment is indicated by

*Figure 2 continued*

an arrow. (B) HeLa cells were treated with control siRNA (siScr) or Capns1 siRNA and then left uninfected or infected with the M90T strain or the *mxiD* mutant for 2 hr. Immunoblot analysis were performed using anti-Capns1, -SUMO1, -SUMO2/3 and -tubulin antibodies. (C) *Capns1*$^{+/+}$ (WT) or *Capns1*$^{-/-}$ (KO) MEFs were uninfected or infected with the M90T strain or the *mxiD* mutant for 2 hr. Immunoblot analysis was performed using anti-SUMO1, -SUMO2/3, and -tubulin antibodies.

DOI: https://doi.org/10.7554/eLife.27444.007

*Shigella* lysates using fluorogenic 7-amino-4-methylcoumarin SUMO substrates (SUMO1-AMC and SUMO2-AMC) (*Figure 4—figure supplement 1* and and *Figure 4—figure supplement 1—source data 1*).

Unlike most proteases that result in extensive degradation of proteins, calpains, which recognize the overall conformation of targeted substrate proteins, usually produces large limited-proteolytic fragments cleaved at the boundary of two domains (*Sorimachi et al., 2012*). The absence of detectable SAE2 proteolytic fragments in immunoblot performed on protein extracts from *Shigella*-infected cells (*Figure 4A–C*) suggests that the epitope of SAE2 recognized by our antibody was further cleaved by calpains. By using another SAE2 antibody, that recognizes a peptide localized around a glutamine in position 421, we were able to detect two different SAE2 proteolytic fragments migrating around 85 kDa and 70 kDa in *Shigella*-infected cells (*Figure 4D*). This result suggests the existence of at least two major calpain cleavage sites on SAE2. We confirmed this finding in vitro by incubating purified recombinant human SAE2 protein with two different doses of purified calpain-1 (which requires micromolar calcium levels for activation) or calpain-2 (which requires millimolar calcium levels for activation) (*Figure 4E*). Incubation of SAE2 with calpain-1 or calpain-2 produced two cleavage products, which are similar in size to those observed in *Shigella*-infected cells. These results identify SAE2 as a novel physiological calpain substrate in vitro and in cells.

## Sumoylation limits *Shigella* invasion and the formation of actin-rich foci

We then evaluated the functional consequences of loss of sumoylation on the pathogenicity of *Shigella* in human epithelial cells depleted for UBC9 using siRNAs and tamoxifen-inducible *Ubc9 KO* MEFs (*Demarque et al., 2011*). In a previous study, we reported that a decrease in SUMO conjugation triggered by SAE2 knockdown favored *Shigella* entry into host cells (*Fritah et al., 2014*). In a similar manner, suppression of UBC9 in HeLa cells led to a significant increase in the number of intracellular *Shigella* (*Figure 5A–B* and *Figure 5—source data 1*). These results correlate with an increase in *Shigella*-induced actin-rich foci, corresponding to the bacteria entry sites (*Figure 5C–D* and *Figure 5—source data 1*). A similar increase in *Shigella* entry and actin polymerization was observed in *Ubc9 KO* MEFs when compared to their wild-type counterparts (*Figure 5E–H* and *Figure 5—source data 1*). Of note, no noticeable alteration of the actin cytoskeleton could be detected in sumoylation-deficient cells in the absence of *Shigella* infection (*Figure 5—figure supplement 1A–B*). Thus, lowering host cell sumoylation facilitates *Shigella*-induced cytoskeletal rearrangements and bacterial uptake into host cells.

## Sumoylation of RhoGDIα regulates *Shigella* entry

To gain molecular insight into how hyposumoylation favors *Shigella* infection, we looked for putative relevant SUMO susbtrates among cytosolic sumoylated proteins identified in previous proteome-wide studies (*Fritah et al., 2014*; *Impens et al., 2014*). Cellular invasion by *Shigella* is known to require massive rearrangements of the host actin cytoskeleton. We thus focused on Rho GDP-dissociation inhibitor alpha (RhoGDIα) given its important role in actin cytoskeleton dynamics (*Garcia-Mata et al., 2011*). Small Rho GTPases are known to be key regulators of actin polymerization and RhoGDIs to down-regulate their biological activity. Notably, RhoGDIα can extract Rho GTPases from membranes and keep them in an inactive state in the cytosol away from their sites of action at membranes. Activation of Rho GTPases family members Rac1, Cdc42 and RhoA is required for *Shigella flexneri* entry process into epithelial cells (*Adam et al., 1996*; *Mounier et al., 1999*). Interestingly, sumoylation of RhoGDIα on lysine138 (K138) has been shown to increase its binding activity to Rho GTPases thereby restraining their biological activity (*Yu et al., 2012*). We thus speculated that a

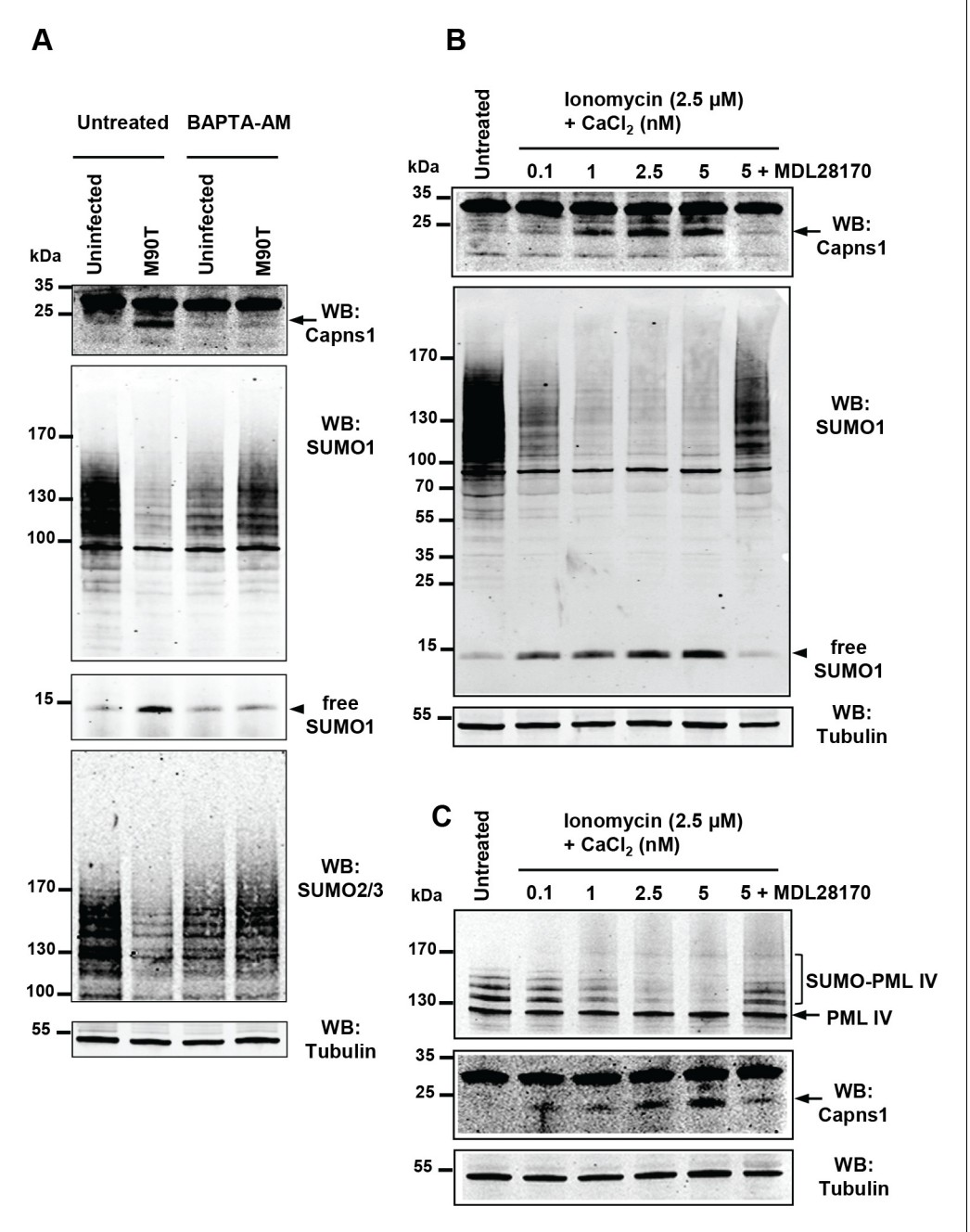

**Figure 3.** Intracellular calcium levels regulate global sumoylation. (**A**) HeLa cells were untreated or pretreated with the calcium-chelating agent BAPTA-AM (10 µM) for 1 hr and then left uninfected or infected with the M90T strain for 2 hr. Immunoblot analysis were performed using anti-Capns1, -SUMO1, -SUMO2/3 and -tubulin antibodies. (**B**) HeLa cells were untreated or treated with the calcium ionophore ionomycin (2.5 µM) and increasing doses of $CaCl_2$(0.1 to 5 nM) for 30 min, with or without addition of the calpain inhibitor MDL28170 (100 µM). Immunoblot analysis were performed using anti-Capns1, -SUMO1 and -tubulin antibodies. The 18 kDa truncated Capns1 fragment is indicated by an arrow. (**C**) HT1080 cells, stably expressing GFP-tagged PML-IV, were untreated or treated with ionomycin (2.5 µM) and increasing doses of $CaCl_2$(0.1 to 5 nM) for 30 min, with or without addition of the calpain inhibitor MDL28170 (100 µM). Immunoblot analysis were performed using anti-GFP, -Capns1 and -tubulin antibodies. The 18 kDa truncated Capns1 fragment is indicated by an arrow.

DOI: https://doi.org/10.7554/eLife.27444.008

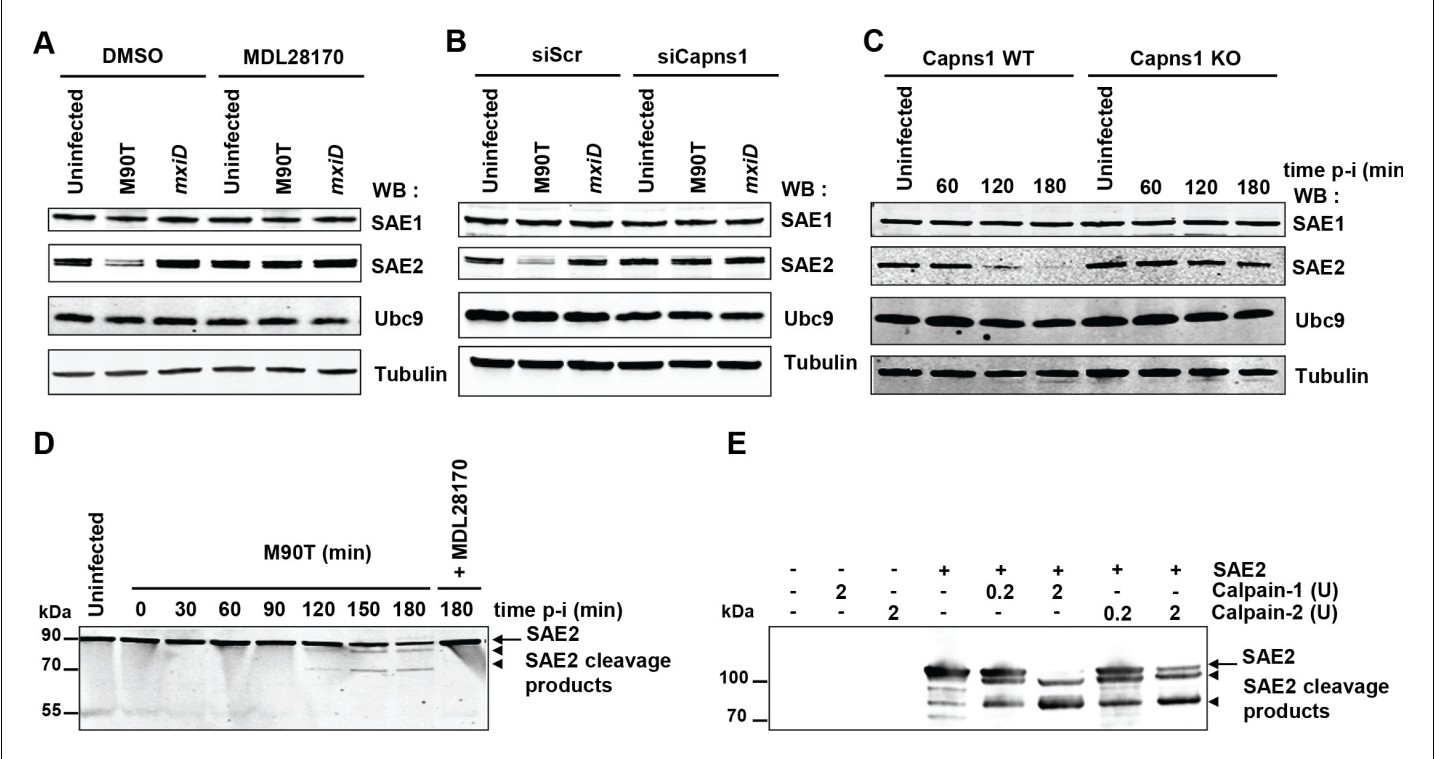

**Figure 4.** SAE2 is a direct calpain substrate. (**A**) HeLa cells were pretreated by vehicle (DMSO) or 100 µM MDL28170 for 1 hr and then left uninfected or infected with the M90T strain or the *mxiD* mutant for 2 hr. Immunoblot analysis were performed using anti-SAE1, -SAE2 (Ab22104), -UBC9 and -tubulin antibodies. (**B**) HeLa cells were treated with control siRNA (siScr) or Capns1 siRNA and then left uninfected or infected with the M90T strain or the *mxiD* mutant for 2 hr. Immunoblot analysis were performed as in C. (**C**) *Capns1^{+/+}* (WT) or *Capns1^{-/-}* (KO) MEFs were uninfected (UI) or infected with *Shigella* for the indicated time. Immunoblot analysis were performed as in C. (**D**) Lanes 1–8: HeLa cells were left uninfected or infected with the M90T strain for the indicated times. Lane 9: HeLa cells were pretreated with 100 µM MDL28170 for 1 hr and then infected with the M90T strain in the presence of 100 µM MDL28170 for 3 hr. Immunoblot analysis was performed using a second anti-SAE2 antibody (D15C11). Arrow indicates full length SAE2 and arrowheads indicate SAE2 cleavage products. (**E**) In vitro proteolysis of SAE2 by calpain-1 or calpain-2 visualized by immunoblotting using SAE2 antibody (D15C11). Recombinant SAE2 was incubated with two different concentrations (0.2U or 2U) of calpain-1 or −2 at 30°C for 20 min. Arrow indicates full length SAE2 and arrowheads indicate SAE2 cleavage products.

DOI: https://doi.org/10.7554/eLife.27444.009

The following source data and figure supplements are available for figure 4:

**Figure supplement 1.** Assessment of desumoylating activities in *Shigella* lysates.

DOI: https://doi.org/10.7554/eLife.27444.010

**Figure supplement 1—source data 1.** Source data file relative to *Figure 4—figure supplement 1*.

DOI: https://doi.org/10.7554/eLife.27444.011

decrease in RhoGDIα sumoylation could increase the recruitment of Rho GTPases at plasma membrane, thus facilitating their activation by *Shigella* for its internalization.

As a first step, we analyzed the effect of depleting RhoGDIα on *Shigella* entry and actin polymerization in human epithelial cells. A significant increase in intracellular bacteria was visible 30 min postinfection in RhoGDIα knockdown cells when compared to control cells (*Figure 6A–B* and *Figure 6— source data 1*). This increase (~60%) was comparable to the increase seen in cells impaired for sumoylation (*Figure 5B–F*). In addition, fluorescence microscopy demonstrated an increased number of actin foci in RhoGDIα-depleted cells in comparison to control cells (*Figure 6C*). Hence, suppressing RhoGDIα activity was sufficient to increase the number of *Shigella* entry sites. Next, to study the impact of sumoylation on RhoGDIα activity in *Shigella* pathogenesis, we compared the effect of wild-type RhoGDIα to that of the corresponding SUMO-deficient mutant (RhoGDIα K138R) (*Yu et al., 2012*) on actin foci formation. Of note, this sumoylation site, that obbeys the canonical consensus motif ψ–K–X–E (where ψ is a large hydrophobic residue), is evolutionarily conserved, suggesting that it may be an important feature of RhoGDIα regulatory function (*Figure 6—figure*

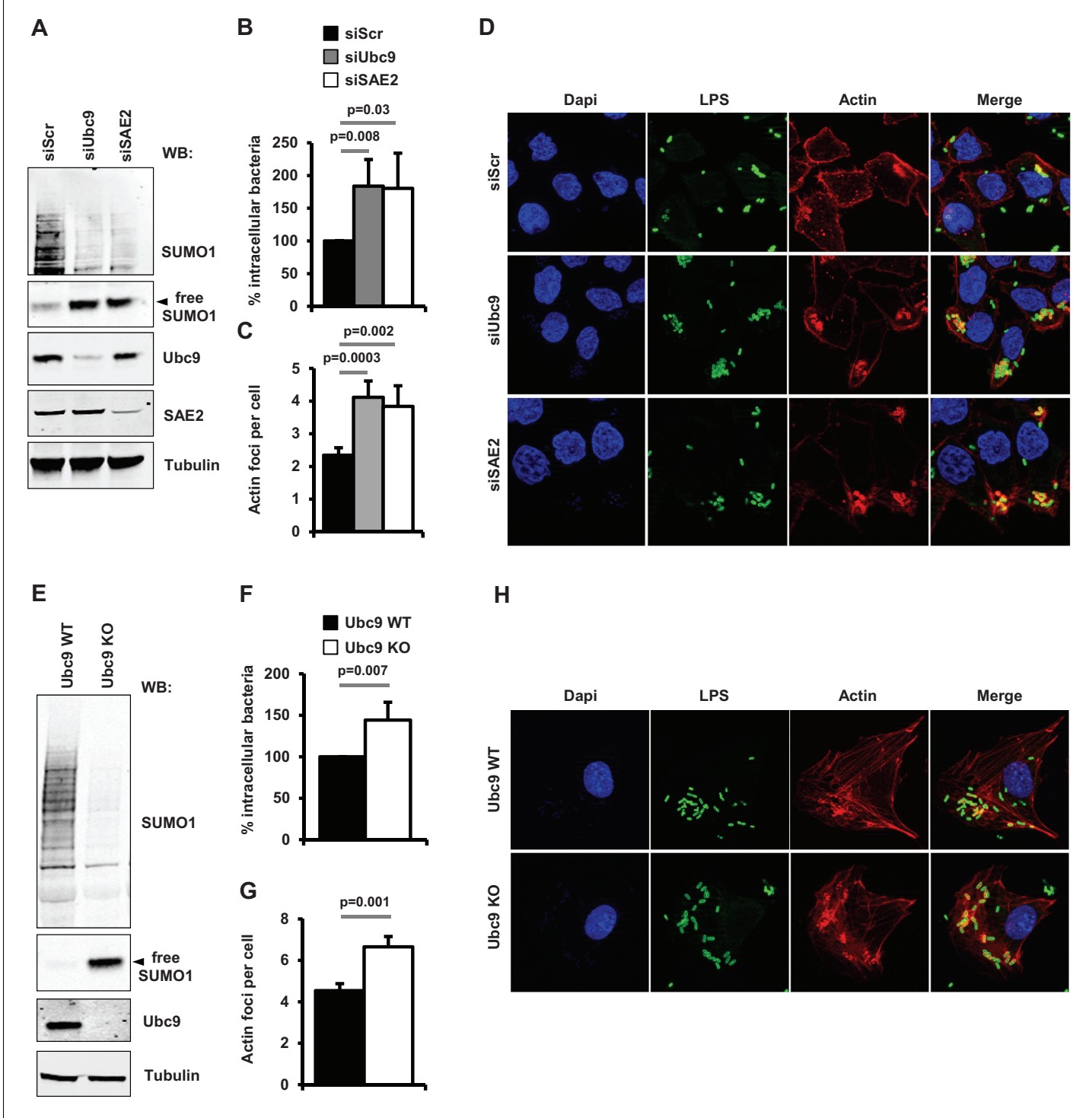

**Figure 5.** Impact of sumoylation on *Shigella* infection. (**A**) HeLa cells were treated with siRNAs for UBC9 and SAE2 or a control siRNA (siScr). Immunoblot analysis were performed using anti-SUMO1, -UBC9, -SAE2 and -tubulin antibodies. (**B**) Percentage of *Shigella* internalization upon siRNA-mediated knockdown of UBC9 and SAE2 in HeLa cells relative to control siRNA. Quantification was performed using the gentamicin protection assay (taken siScr value as 100%) 30 min post-infection. Each value is the mean of six independent experiments ± SEM. (**C**) Actin foci formation upon siRNA-mediated knockdown of UBC9 and SAE2 in HeLa cells. Samples were fixed and processed for actin staining 10 min post-infection. The average number of actin foci per cell ±s.d. is indicated (n = 4, at least 50 cells counted per condition). (**D**) Representative images of *Shigella*-induced actin foci in siRNA-treated HeLa cells after 10 min infection. Samples were processed for bacterial LPS (green), actin (red) and nuclei (blue) staining. (**E**) Primary MEFs from *Ubc9*^+/+^/T2 (WT) and *Ubc9*^fl/-^/T2 (KO) mice (**Demarque et al., 2011**) were treated for 5 days with 4-hydroxy-tamoxifen. Levels of global sumoylation

*Figure 5 continued on next page*

*Figure 5 continued*

and UBC9 were assessed by immunoblot analysis. Tubulin was used as control. (**F**) Percentage of *Shigella* internalization in *Ubc9* WT and *Ubc9* KO MEFs 30 min post-infection. Quantification is as in B (taken *Ubc9* WT value as 100%). Each value is the mean of nine independent experiments ± SEM. (**G**) Actin foci formation in *Ubc9* WT and *Ubc9* KO MEFS. Samples were fixed and processed for actin staining 10 min post-infection. The average number of actin foci per cell ±s.d. is indicated (n = 3, at least 50 cells counted per condition). (**H**) Representative images of *Shigella*-induced actin foci in *Ubc9* WT or *Ubc9* KO MEFs 10 min post-infection. Samples were processed for bacterial LPS (green), actin (red) and nuclei (blue) staining.

DOI: https://doi.org/10.7554/eLife.27444.012

The following source data and figure supplement are available for figure 5:

**Source data 1.** Percentage of *Shigella* internalization and quantification of the average number of actin foci in cells impaired for sumoylation (*Figure 5B, (C,F and G)*).

DOI: https://doi.org/10.7554/eLife.27444.014

**Figure supplement 1.** SUMO loss does not lead to marked alteration of the actin cytoskeleton.

DOI: https://doi.org/10.7554/eLife.27444.013

supplement 1). We took advantage of the fact that both RhoGDIα plasmids lack the 3'-UTR sequence to design a 3'-UTR-targeting siRNA that silenced only endogenous RhoGDIα. Restoration experiments using either GFP-tagged wild-type RhoGDIα or K138R mutant were performed in HeLa cells knockdown for endogenous RhoGDIα (*Figure 6D*). Although the knockdown efficiency of the 3'-UTR-targeting siRNA was not total (~60% knockdown), significantly higher number of *Shigella*-induced actin foci was observed by fluorescence microscopy in cells expressing RhoGDIα K138R, compared to that observed in cells expressing the wild type form (*Figure 6E–F* and *Figure 6—source data 1*). This result demonstrates that impaired sumoylation of RhoGDIα favors *Shigella* entry and could thus contribute to the increased infectivity observed in sumoylation-deficient cells.

To next assess whether sumoylation inhibition impacts the RhoGDIα inhibitory function on Rho GTPase activity, we analyzed the membrane localization of three Rho GTPases (RhoA, Cdc42 and Rac1) in non-infected *Ubc9 WT* and *Ubc9 KO* MEFs. It has been shown that depletion of RhoGDIα, although it decreases the protein levels of the Rho GTPAses through a destabilization process, it significantly increases the proportion of membrane-bound active Rho GTPases (*Boulter et al., 2010*). Enrichment in the cell membrane fraction using ultracentrifugation revealed that, despite similar amounts of GTPases in total protein extracts, a modest yet significant increase in the amount of membrane-bound Rho GTPases was visible in sumoylation-deficient MEFs, relative to wild-type MEFs (*Figure 6G–H* and *Figure 6—source data 1*). These results indicate that loss of sumoylation favors accumulation of Rho GTPases at the plasma membrane, thus providing a supportive environment for *Shigella* entry into host cell.

## Endogenous RhoGDIα and SUMO proteins are recruited at bacteria entry sites and *Shigella* rapidly impairs RhoGDIα sumoylation

The RhoGDI–Rho GTPase complex has been shown to shuttle between the cytosol and the membrane (*Garcia-Mata et al., 2011*). To thus visualize the potential recruitment of engodenous SUMO to the sites of bacterial entry, we performed immunofluorescence experiments in *Shigella*-infected *Ubc9 WT* MEFs using SUMO1 and SUMO2/3 antibodies. A consistant enrichment of both SUMO1 and SUMO2/3 was observed at the actin-rich foci soon after infection (*Figure 7A–D*, upper panels and black bars, and *Figure 7—source data 1*). Similar findings were obtained in infected HeLa cells (*Figure 7—figure supplement 1*). Immunostaining for SUMO1 and SUMO2/3 performed in *Ubc9 KO* MEFs revealed a significant decrease in SUMO signal at the *Shigella* entry sites. This finding indicates that the presence of SUMO at actin foci in *Ubc9 WT* MEFs corresponds to SUMO-conjugated proteins and not to free SUMO1 or SUMO2/3 (*Figure 7A–D*, lower panels and white bars). We then looked at the distribution of RhoGDIα upon *Shigella* infection. This revealed a noticable enrichment at actin-rich foci in *Ubc9 WT* MEFs. Such an accumulation is still observable in the sumoylation-deficient *Ubc9 KO* MEFs, suggesting that sumoylation does not impact RhoGDIα recruitment to plasma membrane (*Figure 7E–F* and *Figure 7—source data 1*).

Finally, we assessed whether the global loss of SUMO conjugates induced by *Shigella* infection translates into a similar decrease in the sumoylation state of endogenous RhoGDIα. Immunoprecipitation followed by western blotting in uninfected cells revealed, in addition to the unmodified ~25 kDa RhoGDIα, a ~40 kDa RhoGDIα species, which was detected by two different anti-SUMO1

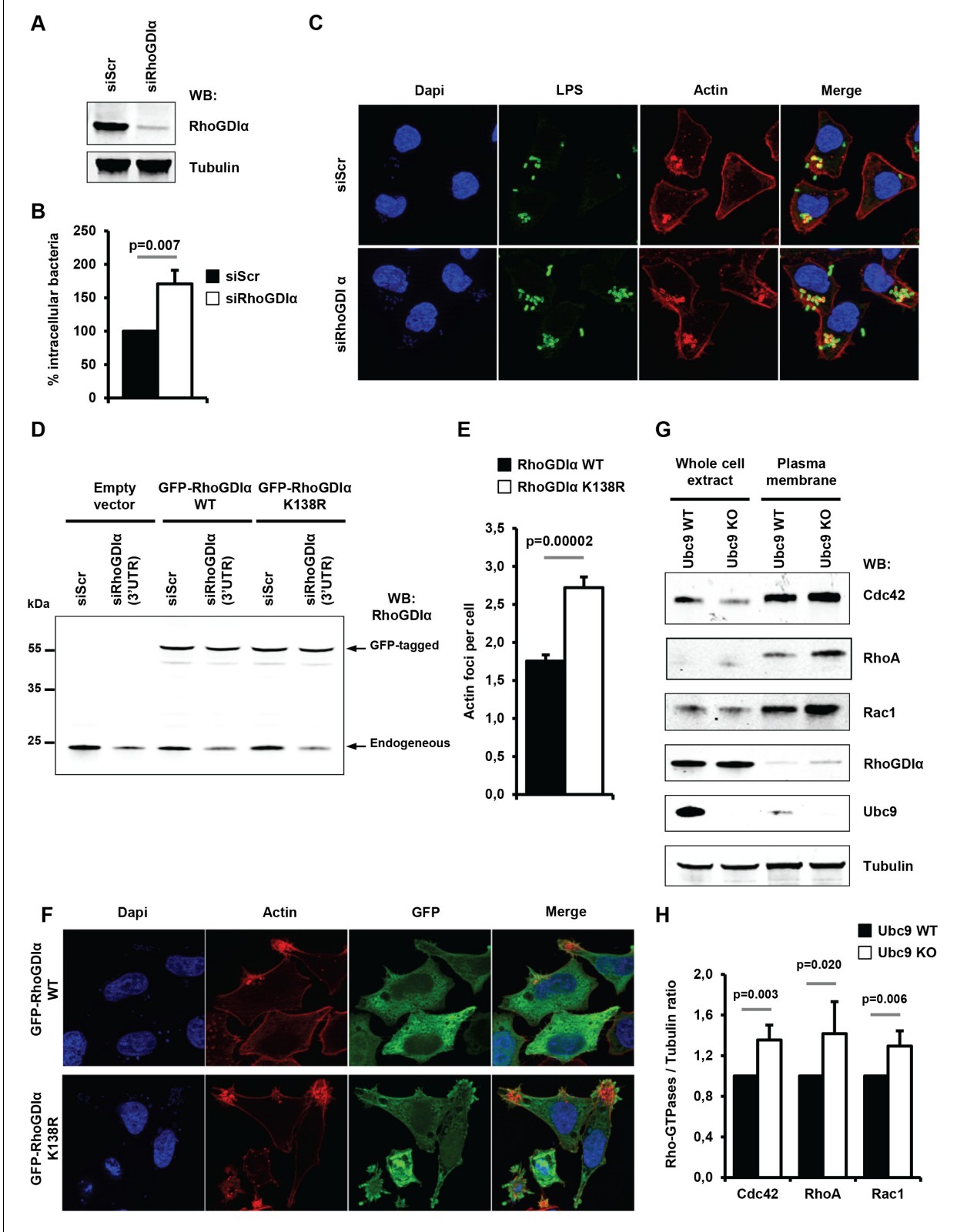

**Figure 6.** Sumoylation of RhoGDIα regulates *Shigella* internalization in epithelial cells. (**A**) HeLa cells were treated with siRNA for RhoGDIα or a control siRNA (siScr). Immunoblot analysis was performed using anti-RhoGDIα and -tubulin antibodies. (**B**) Percentage of *Shigella* internalization upon siRNA-mediated knockdown of RhoGDIα relative to control siRNA. The percentage of internalized bacteria was quantified using the gentamicin protection assay (taken siScr value as 100%) 30 min post-infection. Each value is the mean of eleven independent experiments ± SEM. (**C**) Representative images of
*Figure 6 continued on next page*

*Figure 6 continued*

*Shigella*-induced actin foci in siRNA-treated HeLa cells after 10 min infection. Samples were processed for bacterial LPS (green), actin (red) and nuclei (blue) staining. (D) Hela cells co-transfected with siRhoGDIα (targeting the 3'UTR) together with either GFP-tagged sRhoGDIα WT or GFP-tagged RhoGDIα K138R. Immunoblotting was performed using a RhoGDIα antibody. Arrows indicate GFP-tagged and endogenous RhoGDIα proteins. (E) Hela cells co-transfected as in D and infected with M90T for 10 min. Samples were fixed and processed for actin staining 10 min post-infection. The average number of actin foci per cell ±s.d. is indicated (n = 4, at least 50 cells counted per condition). (F) Representative images of *Shigella*-induced actin foci in HeLa cells co-transfected as in D with GFP-tagged RhoGDIα constructs (green) after 10 min infection. Samples were processed for actin (red) and nuclei (blue) staining. (G) Immunoblot analysis was performed on whole-cell lysates or plasma membrane fractions (recovered by ultracentrifugation) from *Ubc9* WT or *Ubc9* KO MEFs using anti-Cdc42, -RhoA, -Rac1, - RhoGDIα, -UBC9 and -tubulin antibodies. (H) Quantification of the immunoblot signals obtained from *Ubc9* WT or *Ubc9* KO MEF protein extracts are presented as RhoGTPase signal (Cdc42, RhoA or Rac1, as indicated on the x-axis) relative to tubulin signal (mean of five independent experiments ± s.d.).

DOI: https://doi.org/10.7554/eLife.27444.015

The following source data and figure supplement are available for figure 6:

**Source data 1.** Source data file relative to *Figure 6B, E and H*.

DOI: https://doi.org/10.7554/eLife.27444.017

**Figure supplement 1.** Conservation of the sumoylation site within RhoGDIα amino acid sequence in various species.

DOI: https://doi.org/10.7554/eLife.27444.016

antibodies (Y299 and 21C7 clones), indicating SUMO-modified RhoGDIα (*Figure 8*). We then investigated the impact of *Shigella* infection on RhoGDIα sumoylation with time. Whereas the levels of unmodified RhoGDIα remained unaffected, we observed the complete disappearance of SUMO-RhoGDIα as quickly as 30 min post-infection (*Figure 8*). The reduction in the global amount of SUMO1 conjugates was only moderate in these conditions (*Figure 1* and *Figure 1—figure supplement 1B*), indicating that RhoGDIα is highly sensitive to sumoylation inhibition. Thus, RhoGDIα is a *bona fide* SUMO substrate in vivo and SUMO-modified RhoGDIα is rapidly lost upon *Shigella* infection.

## Discussion

Pathogenic organisms possess the remarkable ability to exploit post-translational modification mechanisms to modulate host factors for their own survival and propagation. Whereas some bacterial pathogens have been shown to alter the sumoylation of host proteins, the mechanisms through which the bacteria interfere with the SUMO machinery and the identity of the SUMO targets remain largely undefined. In this study, we show that *Shigella* induces a massive decrease in SUMO1 and SUMO2/3 conjugates at late time post-infection in epithelial cells in culture and in the intestinal mucosa. This global loss in sumoylation relies on activation of calpain proteases that target the SUMO E1 enzyme SAE2 for degradation, thus leading to sumoylation inhibition. In addition, we show that SUMO-modified RhoGDIα is rapidly lost upon *Shigella* infection favoring cytoskeletal rearrangements and bacterial entry. To our knowledge, this is the first characterization of a SUMO substrate targeted by bacteria to enhance infectivity. Thus, in addition to identifying sumoylation of RhoGDIα as an important event countering cytoskeletal remodeling and bacterial entry, our work reveals the ability of calcium signals to control global sumoylation levels.

Calpain proteases constitute a family of calcium-dependent cysteine proteases involved in a wide range of cellular functions, including cytoskeletal rearrangements, apoptosis and cell survival (*Ono and Sorimachi, 2012*). An increase in free intracellular calcium is required to induce the calpain conformational changes necessary for their activity and substrate recognition. Upon host cell invasion, *Shigella* induces both local and global calcium responses. Whereas the local calcium response at *Shigella* entry sites peaks at 15 min post-infection, a global increase in calcium signaling is observed shortly thereafter (*Bonnet and Tran Van Nhieu, 2016*). Local elevation of intracellular calcium levels at early stage leads to calpain activation that affects the dynamics of cytoskeletal reorganization to promote *Shigella* invasion. At later stage, global calcium responses associated with sustained calpain activation leads to slow necrotic cell death (*Bergounioux et al., 2012*). Our findings that inhibiting either intracellular calcium influx or calpain activity prevented *Shigella*-induced loss of SUMO-conjugates and, conversely, that the sole treatment with calcium and ionomycin in the absence of *Shigella* triggered sumoylation inhibition indicate that increased cytosolic calcium and

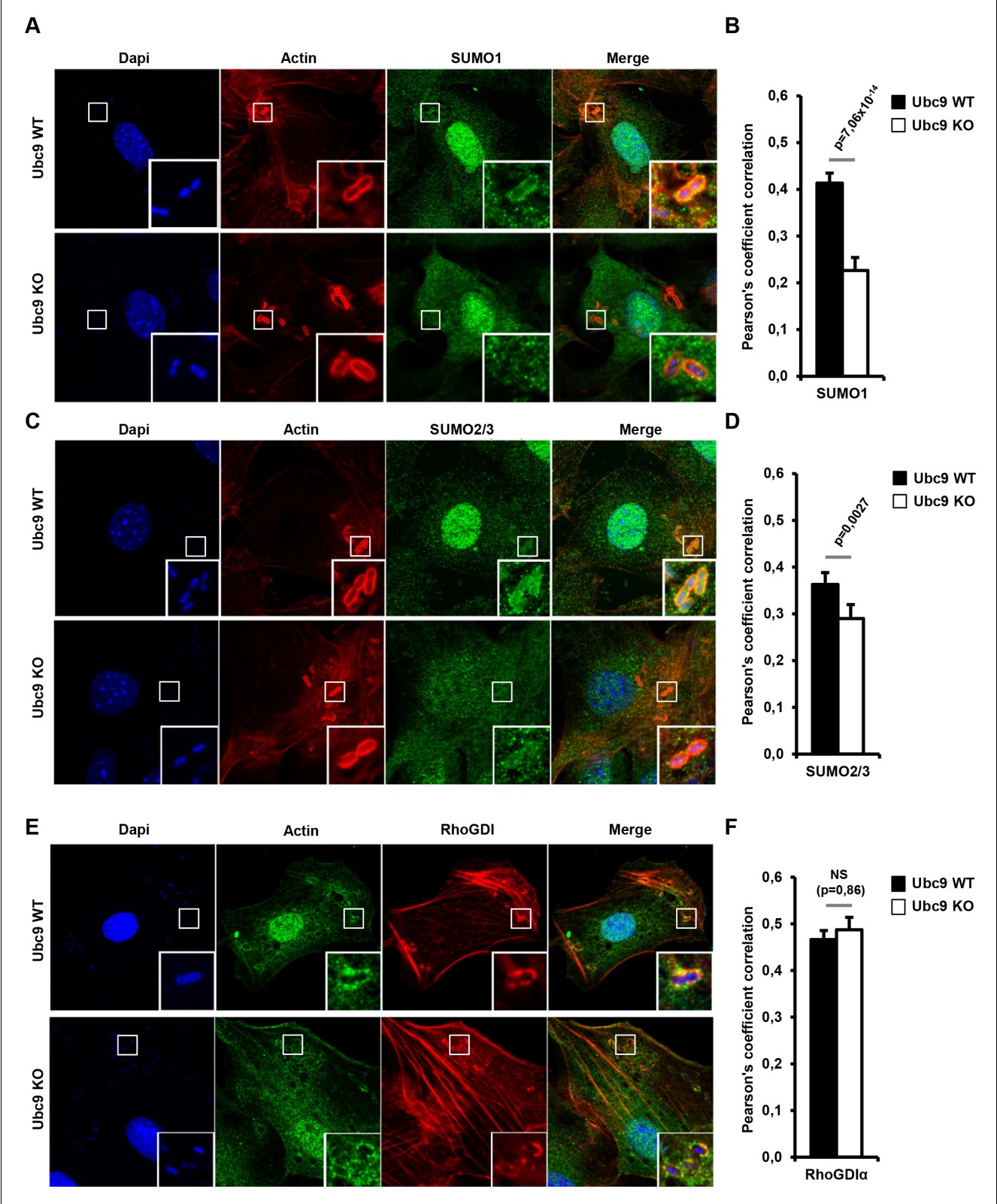

**Figure 7.** Endogenous RhoGDIα and SUMO are localized at *Shigella*-induced actin foci. (**A**) SUMO1 accumulates at *Shigella* (M90T) entry sites. Representative ApoTome-generated micrographs of *Shigella*-infected *Ubc9* WT or *Ubc9* KO MEFs after 10 min infection. Samples were fixed and processed for immunostaining using anti-SUMO1 antibody (green) and staining of actin (red) and nuclei (blue) (white square, inset). (**B**) The Pearson's coefficient (Rr) was used to measure the signal intensity correlation between SUMO1 and *Shigella*-induced actin foci stainings. Data are means ±SEM (at

*Figure 7 continued on next page*

*Figure 7 continued*

least 40 foci analyzed per condition). p value calculated as described in Materials and Methods. (**C**) SUMO2/3 accumulates at *Shigella* (M90T) entry sites. Same as in A using a SUMO2/3 antibody. (**D**) Same as in B with SUMO2/3 signal. (**E**) Recruitment of RhoGDIα is at *Shigella* (M90T) entry sites. Same as in A using a RhoGDIα antibody. (**F**) Same as in B with RhoGDIα signal. NS: non significant.

DOI: https://doi.org/10.7554/eLife.27444.018

The following source data and figure supplement are available for figure 7:

**Source data 1.** Source data files relative to *Figure 7B, D and F*.

DOI: https://doi.org/10.7554/eLife.27444.020

**Figure supplement 1.** SUMO1 and SUMO2/3 accumulate at *Shigella* (M90T) entry site in Hela cells.

DOI: https://doi.org/10.7554/eLife.27444.019

subsequent calpain activation are responsible for SAE2 degradation and impairment of sumoylation. To our knowledge, a single study describing a putative role of calpains in the modulation of specific sumoylation events has been reported. In this work, forced expression of calpain three was shown to lead to the cleavage the SUMO E3 ligase PIAS3 and subsequent inhibition of its enzymatic activity (*de Morrée et al., 2010*). A recent study reported that HeLa cells infected by *Shigella* show reduced sumoylation together with a slight decrease in UBC9 protein levels (*Sidik et al., 2015*). We repeatedly failed to detect any decrease in the amount of UBC9 following *Shigella* infection whereas SAE2 was systematically found to be lost. The reason for this discrepancy remains unknown. Using a panel of *Shigella* strains mutated for a series of bacterial effectors, we show that mutants lacking the ability to elicit a stress response at the plasma membrane, as detected by actin foci formation, are no longer able to trigger loss of SUMO conjugates. Calpain proteases have been shown to be activated by plasma membrane injuries in order to contribute to membrane repair (*Godell et al., 1997*; *Mellgren et al., 2009*; *Mellgren et al., 2007*). One may thus hypothesize that, by triggering pore formation and subsequent plasma membrane stress, *Shigella* infection induces a calpain-dependent loss of sumoylation.

Current knowledge of the mechanisms through which bacteria interfere with the SUMO enzymatic machinery remains largely limited. To date, a unique example of a pathogen protein targeting the SAE1/SAE2 heterodimer is the adenoviral protein Gam1 that recruits SAE1/SAE2 to the Cullin2/5-

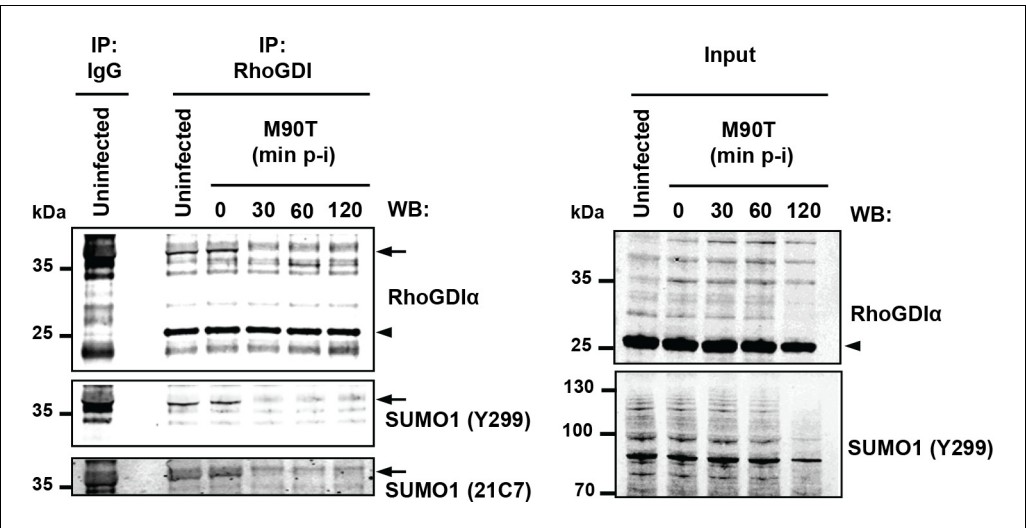

**Figure 8.** *Shigella* infection leads to loss in SUMO- RhoGDIα. Endogenous RhoGDIα is modified by SUMO1 and rapidly desumoylated upon *Shigella* infection. Whole-cell extracts harvested from uninfected or *Shigella* (M90T)-infected Hela cells (from 0 to 120 min post-infection) were subjected to immunoprecipitation (IP) using anti-RhoGDIα or control IgG. Immunoprecipitates (left panel) and input lysates (right panel) were analysed by immunoblot analysis using anti-RhoGDIα, and -SUMO1 (clones Y299 and 21C7) antibodies. Arrowhead indicates unmodified RhoGDIα and arrow indicates SUMO1- RhoGDIα.

DOI: https://doi.org/10.7554/eLife.27444.021

EloB/C-Roc1 ubiquitin ligase complex, thus leading to SAE1 ubiquitin-dependent degradation (*Boggio et al., 2007*). By contrast, the majority of bacteria known to interfere with sumoylation have been shown to target the E2 conjugating enzyme UBC9 (*Srikanth and Verma, 2017*). *Salmonella* Typhimurium depletes UBC9 in infected cells by upregulating the expression of two microRNAs (miR30c and miR30e) that affect UBC9 transcript stability (*Verma et al., 2015*). Another example is the Gram-positive bacteria *Listeria monocytogenes* that leads to decreased sumoylation together with UBC9 degradation. Whereas MG132, that inhibits proteasomal degradation but also calpain activity (*Lee and Goldberg, 1998*), partially restored the profile of sumoylated proteins in infected cells, it failed to prevent UBC9 degradation (*Ribet et al., 2010*). The mechanisms underlying these two apparently paradoxical findings remain to be identified. The secretion of the pore-forming toxin (PFT) LLO triggers UBC9 degradation and this bacterial toxin alone can recapitulate the decrease in sumoylation observed during *Listeria* infection. Moreover, this effect on host cells has also been observed for PFTs from other Gram-positive bacteria, such as PFO and PLY (*Ribet et al., 2010*). It is possible that, upon *Shigella* infection, the translocator-forming IpaB and IpaC proteins, by inducing a stress at the plasma membrane, act in a similar manner. In this context, it will be interesting to investigate whether the abilities of the Gram-positive bacteria PFTs to decrease host sumoylation might be linked to calpain activation since many PFTs, including LLO, are described to activate these proteases by elevating intracellular calcium levels (*Bischofberger et al., 2012*).

A common theme in the pathogenicity of bacteria is the manipulation of host cells by targeting the cytoskeleton for their own needs (*Barbieri et al., 2002*). Among the multiple regulation steps of the actin cytoskeleton, bacterial factors interfere preferentially with Rho GTPases either directly *via* covalent modification or through interfacing with regulators of Rho GTPase control. As Rho GTPases are active in the GTP-bound state, several bacteria produce toxins that modulate the nucleotide state of the Rho GTPases for activation or inhibition (*Finlay, 2005*). For example, *Shigella* injects into host cells the virulence factors IpgB1 and IpgB2 that activate the Rho GTPases Rac1, Cdc42 and RhoA through their guanine nucleotide exchange factor activity toward these proteins (*Klink et al., 2010*; *Ohya et al., 2005*). However, few reports have described direct effects of bacterial infection on RhoGDIα activity. One example is the *Yersinia* effector YpkA that mimics eukaryotic RhoGDIα, leading to global Rho GTPase inhibition and cytoskeletal disruption (*Prehna et al., 2006*). We report here that RhoGDIα silencing increases the number of intracellular *Shigella* and actin foci, indicating that RhoGDIα regulatory functions are required to limit *Shigella* entry into host cells. Moreover, sumoylation of RhoGDIα is important for this activity as a SUMO-deficient RhoGDIα mutant shows a reduced ability to impair bacterial entry. These data are consistent with the finding that SUMO-RhoGDIα plays an inhibitory role on actin polymerization (*Yu et al., 2012*). It is thus tempting to speculate that the rapid loss in SUMO-RhoGDIα triggered by *Shigella* entry could promote de novo infection by extracellular *Shigella* thus amplifying the infection process. Intriguingly, whereas the degradation of SAE2 is hardly visible before 2 hr post-infection, the decrease in SUMO-RhoGDI-α occurs earlier, as soon as 30 min following infection, and coincides with the first visible signs of global hyposumoylation. Since the loss of SAE2 and SUMO conjugates are both calpain-dependent, a possible explanation for these differing kinetics may be that the two events result from the two consecutive waves of calcium responses triggered by *Shigella* infection. Whereas, the second, global calcium response would trigger complete SAE2 degradation, ultimately leading to a generalized loss of SUMO-conjugated proteins, the rapid loss of SUMO-RhoGDIα would most likely result from the first and localized wave of Ca2+ responses induced by *Shigella* in the vicinity of the entry sites. This response would lead to calpain-induced degradation of a small local pull of SAE2, barely detectable by Western blot, and subsequent loss of sumoylated RhoGDIα. Such a local and long-lasting Ca2+ response was shown previously to be confined to bacterial invasion sites, being induced as early as 5 min after bacterial contact with epithelial cells, with a peak response at 15 min (*Bonnet and Tran Van Nhieu, 2016*). Moreover, these local calcium responses at Shigella entry sites occur at about 10 μM, a concentration quite compatible with the activation of calpains (*Khorchid and Ikura, 2002*). In line with this notion, local calpain activation at the plasma membrane has been extensively described for many membrane-associated substrates, such as FAK, talin, insulin receptor and VE-cadherin (*Chang et al., 2017*; *Su and Kowalczyk, 2017*; *Yuasa et al., 2016*). If, as we surmise, sumoylation of RhoGDIα were indeed to take place in the vicinity of the bacterial entry sites, it is thus likely that the local activation of calpains by an initially localized rise in calcium may lead to SAE2 cleavage and subsequent loss of SUMO-RhoGDIα - and potentially of a limited pool of other SUMO substrates - at

*Shigella* invasion sites. A more definitive clarification of this issue must, however, await the development of probes permitting the visualization of localized sumoylation dynamics.

In conclusion, the data presented here describe a novel mechanism by which *Shigella* promotes its own infection capacity by rapidly decreasing the sumoylation state of RhoGDIα. It remains to be determined to which extent the sumoylation of other host substrates can contribute to limit *Shigella* invasion. *Shigella,* however, can also positively modulate sumoylation of a restricted set of substrates. For example, the *Shigella* effector OspF has been described to be sumoylated, favoring its translocation into the nucleus (*Jo et al., 2017*). Moreover, using a proteomic approach, we found that, whereas *Shigella* mainly induces hyposumoylation at early stage of infection, a small number of cellular substrates also become hypersumoylated (*Fritah et al., 2014*). Clearly, full dissection of the spatio-temporal interplay between *Shigella* and sumoylation will require further investigation.

In addition, our work reveals a previously unknown strategy for modulating the global levels of cellular sumoylation through a calcium/calpain-dependent process that may have important implications in a number of pathological or physiological situations. Calcium signaling is involved in a multitude of biological processes, such as synaptic function, muscle contraction and cardiac activity (*Clapham, 2007*). It is thus not surprising that alteration in calcium homeostasis is known to participate in a number of pathological processes including cardiovascular diseases, neurological disorders and cancer (*Carafoli, 2004*). An interesting challenge in the future will be to probe whether situations associated with changes in cytosolic calcium levels, such as during the sleep-wake cycle (*Berridge, 2014*), could translate into the modulation of global cellular sumoylation. Morever, the highly localized nature of calcium signals, as exemplified by the local calcium response confined to the *Shigella* invasion site (*Bonnet and Tran Van Nhieu, 2016*), offers the intriguing possibility for localized variations in sumoylation, as shown recently for ubiquitination (*McGourty et al., 2016*).

# Materials and methods

**Key resources table**

| Reagent type (species) or resource | Designation | Source or reference | Identifiers |
|---|---|---|---|
| strain, strain background (Shigella flexneri serotype 5a) | M90T strain | PMID: 6279518 | Taxonomy ID: 1086030 |
| strain, strain background (Shigella flexneri serotype 5a | *mxiD* | PMID: 8437520 | N/A |
| strain, strain background (Shigella flexneri serotype 5a) | mxiE | PMID: 12142411 | N/A |
| strain, strain background (Shigella flexneri serotype 5a) | ospG | PMID: 16162672 | N/A |
| strain, strain background (Shigella flexneri serotype 5a) | ipaB | PMID: 1582426 | N/A |
| strain, strain background (Shigella flexneri serotype 5a) | ipaC | PMID: 19165331 | N/A |
| strain, strain background (Shigella flexneri serotype 5a) | ipaC/pC57 | PMID: 19165331 | N/A |
| strain, strain background (Shigella flexneri serotype 5a) | ipaC/pC351 | PMID: 19165331 | N/A |
| strain, strain background (Shigella flexneri serotype 5a) | virA | PMID: 22423964 | N/A |
| strain, strain background (Shigella flexneri serotype 5a) | ipgD | PMID: 8478058 | N/A |
| cell line (Hela) | CCL-2 | ATCC | ATCC CCL2/CVCL_0030 |
| cell line (HT1080) | CCL-121 | ATCC | ATCC CCL-121/CVCL_0317 |
| cell line (HT1080) | GFP-PML-IV | PMID: 23530056 | N/A |
| cell line (Hela) | TAP-SUMO1 | PMID: 25097252 | N/A |
| cell line (Hela) | TAP-SUMO2 | PMID: 25097252 | N/A |

*Continued on next page*

*Continued*

| Reagent type (species) or resource | Designation | Source or reference | Identifiers |
|---|---|---|---|
| cell line (HT1080) | UBC9-auxin degron | this paper | N/A |
| genetic reagent (siRNA) | control | Dharmacon | #D-001810–10 |
| genetic reagent (siRNA) | Capns1 | Dharmacon | #L-009979–00 |
| genetic reagent (siRNA) | Ubc9 | Dharmacon | #L-004910–00 |
| genetic reagent (siRNA) | SAE2 | Dharmacon | #L-005248–01 |
| genetic reagent (siRNA) | RhoGDIα | Dharmacon | #L-016253–00 |
| transfected construct (Plasmid) | pEGFP-RhoGDIα WT | PMID: 22393046 | N/A |
| transfected construct (Plasmid) | pEGFP-RhoGDIα K138R | PMID: 22393046 | N/A |
| antibody | anti-SAE1 | Abcam | #ab97523/AB_10681015 |
| antibody | anti-SAE2 | Abcam | #ab22104/AB_446785 |
| antibody | anti-SUMO1 | Abcam | Y299/AB_778173 |
| antibody | anti-SUMO2/3 | Abcam | 8A2/AB_1658424 |
| antibody | anti-UBC9 | Abcam | EP2938Y/AB_1267373 |
| antibody | anti-SUMO1 | DSHB Iowa | 21C7/AB_2198257 |
| antibody | anti-Calpastatin | Cell Signaling Technology | #4146/AB_2244162 |
| antibody | anti-SAE2 | Cell Signaling Technology | D15C11/AB_10889561 |
| antibody | anti-Cdc42 | Cell Signaling Technology | 11A11/AB_10695738 |
| antibody | anti-RhoA | Cell Signaling Technology | 67B9/AB_10693922 |
| antibody | anti-Tubulin | Cell Signaling Technology | DM1A/AB_1904178 |
| antibody | anti-RanGAP1 | Santa Cruz | C-5/AB_2176987 |
| antibody | anti-GFP | Santa Cruz | C-2 |
| antibody | anti-RhoGDIα | Merck Millipore | #06–730/AB_310229 |
| antibody | anti-Capns1 | Merck Millipore | MAB3083/AB_2070014 |
| antibody | anti-Rac1 | Merck Millipore | 23A8/AB_309712 |
| antibody | anti-LPS | PMID: 25097252 | N/A |
| antibody | anti-SP100 | PMID: 7559785 | N/A |
| chemical compound, drug | Phalloidin–Tetramethylrhodamine B isothiocyanate | Sigma | P1951 / AB_2315148 |
| chemical compound, drug | Dapi | Sigma | D9542 |
| chemical compound, drug | Cytochalasin D | Sigma | C8273 |
| chemical compound, drug | MDL 28170 | Sigma | M6690 |
| chemical compound, drug | Ionomycin | Sigma | I3909 |
| chemical compound, drug | BAPTA-AM | Enzo life sciences | BML-CA411-0025 |
| chemical compound, drug | Indole-3-acetic acid | Sigma | I5148 |
| chemical compound, drug | N-Ethylmaleimide | Sigma | E3876 |
| peptide, recombinant protein | Recombinant SAE2 | Novus biologicals | NBP2-50574-20ug |
| peptide, recombinant protein | Recombinant Calpain-1 | Merck Millipore | 208712 |
| peptide, recombinant protein | GST-SENP2cat | this paper | NP_06760.2 |
| peptide, recombinant protein | SUMO1-AMC | Boston Biochem | UL-551 |
| peptide, recombinant protein | SUMO2-AMC | Boston Biochem | UL-758 |
| software, algorithm | Icy software | Institut Pasteur | PMID: 22743774 |

## Bacterial strains and cell culture

*Shigella flexneri* serotype 5a strains were isolated on congo red agar plates. The invasive wild-type strain M90T, its isogenic non-invasive derivative *mxiD* (impaired for T3SS) and isogenic mutants for

various effectors (*mxiE*, *ospG*, *ipaB*, *ipaC*, *virA* and *ipgD*) were used. Two strains expressing IpaC variants (*ipaC*/pC57 and *ipaC*/pC351) were also used (*Mounier et al., 2009*). For infection experiments, strains were cultured in BTCS medium (Difco) overnight at 37°C with agitation. Subcultures were performed for 3 hr to reach the exponential phase and resuspended in DMEM medium (Invitrogen). Human cell lines HeLa CCL-2 and HT1080 were obtained from ATCC and grown according to the supplier's recommendations. The HT1080 cell line stably expressing GFP-PML IV, generated in our lab, was maintained as previously reported (*Erker et al., 2013*). HeLa cells overexpressing TAP-SUMO-1 and TAP-SUMO-2, kindly provided by Ronald T. Hay (University of Dundee, Scotland, UK), were maintained as previously reported (*Fritah et al., 2014*). None of these cell lines belongs to the list of commonly misidentified cell lines (ICLAC). All cell lines cell has been routinely tested for mycoplasma contamination using the PCR Mycoplasma Test Kit II (PromoKine).

## siRNA and plasmid transfection

Hela cells were transfected for 72 hr using Lullaby reagent (OZ biosciences) with siRNA from Dharmacon against Capns1 (#L-009979–00), UBC9 (#L-004910–00), SAE2 (#L-005248–01), RhoGDIα (#L-016253–00) or control siRNA (#D-001810–10, Dharmacon) according to the manufacturers'instructions. The pEGFP-RhoGDIα WT or pEGFP-RhoGDIα K138R expression vectors were, respectively, a kind gift from Dr. Mark R. Philips (New York University School of Medicine, New York) and Dr. Chuanshu Huang (New York University School of Medicine, New York). HeLa cells were transfected with plasmids by using Lipofectamine 2000 (Invitrogen) according to the manufacturer's protocol.

## Antibodies and reagents

For immunoblotting and immunofluorescence experiments we used the following antibodies. Rabbit polyclonal anti-SAE1 (#ab97523) and anti-SAE2 (#ab22104), rabbit monoclonal anti-SUMO1 (Y299) and anti-UBC9 (EP2938Y) and mouse monoclonal anti-SUMO2/3 (8A2) were purchased from Abcam. Mouse monoclonal anti-SUMO1 (21C7) was from DSHB Iowa. Rabbit polyclonal anti-Calpastatin (#4146), rabbit monoclonal anti-SAE2 (D15C11), anti-Cdc42 (11A11) and anti-RhoA (67B9) and mouse monoclonal anti-Tubulin (DM1A) were purchased from Cell Signaling Technology. Mouse monoclonal anti-RanGAP1 (C-5) and anti-GFP (C-2) were purchased from Santa Cruz. Rabbit polyclonal anti-RhoGDIα (#06–730) and mouse monoclonal anti-Capns1 (MAB3083) and anti-Rac1 (23A8) were purchased from Merck Millipore. Rabbit polyclonal antibodies to *Shigella flexneri* 5a LPS and SP100 (*Carvalho et al., 1995*) are home made. TRITC-labelled phalloidin for vizualization of actin cytoskeleton in mammalian cells, Dapi for labelling nuclei and Cytochalasin D for blocking actin cytoskeleton were purchased from Sigma. The calpain inhibitor MDL28170 (Z-Val-Phe-aldehyde) (Sigma) was added at a 100 mM final concentration to the culture medium. Cytochalasin D (Sigma) was added at 15 min prior to infection (5 µM). The calcium ionophore ionomycin (Sigma) and $CaCl_2$ (Sigma) were used at indicated doses as calpains inducers. BAPTA-AM (10 µM, Enzo Life Sciences) was used as a cell permeant Ca2+ chelator to inhibit intracellular calpains activity.

## Invasion assays

Bacterial invasion of human cells and MEFs was performed using gentamycin protection assay (*Lapaquette et al., 2010*). Epithelial cell monolayers were infected with the indicated moi (multiplicity of infection). After 10 min of centrifugation at 1000 g and a 10 min incubation period at 37°C (5% $CO_2$), the infected cells were washed twice with PBS, and fresh cell culture medium containing 50 mg/mL of gentamicin was added for 1 hr. To determine the number of intracellular bacteria, the cell monolayer was washed twice with PBS and lysed with 1% Triton X-100 (Sigma) in PBS, then mixed, diluted and plated onto TCS agar plates to determine the number of colony forming unit (CFU) recovered from the lysed monolayer.

## Auxin-inducible impaired sumoylation

HT1080 cells, stably expressing UBC9 fused to an auxin-inducible degron were used (manuscript in preparation). Degradation of the expressed UBC9-auxin degron fusion, leading to impaired sumoylation, was induced by adding auxin, indole-3-acetic acid (Sigma; 200 µM final), for 24 hr to the cell culture medium (DMEM + Glutamax, Gibco).

## Immunofluorescent staining

After bacterial infection, cells were fixed with 4% paraformaldehyde (PFA) and immunostained overnight at 4°C, with the indicated specific primary antibodies. A 1 hr incubation with secondary antibodies and/or TRITC-labelled phalloidin was performed at room temperature. To determine the number of actin foci per cell, at least 100 *Shigella*-infected cells were counted. Each microscopy image is representative of at least three independent experiments. Intestinal samples from newborn mice were fixed for 2 hr in 4% PFA and kept in 70% ethanol before paraffin embedding. Microtome sections of 7 μm were prepared. Sections were rehydrated, permeabilized with 0.1% Triton X-100 for 15 min, saturated with Ultra-V-Block (Thermo Scientific) and then incubated overnight at 4°C with SUMO2/3 antibody. After washing, the sections were incubated with a goat anti-rabbit Cy3-conjugated secondary antibody (Jackson Immunotech) for 1 hr at room temperature. Nuclei were counternstained with DAPI, and slides were mounted (Prolong, Life technologies). All images were acquired using Apotome microscope (Zeiss). Pearson's correlation coefficient (Rr) was used as a measure of the co-localization and calculated using the colocalization studio plugin of Icy software (*Lagache et al., 2015*).

## Immunoblot analysis

Whole-cell protein extracts were prepared from cell monolayer by adding directly 2x Laemmli sample buffer (Bio-rad). Newborn mice gut was dissected and homogenized in 1 mL of lysis buffer (50 mM Tris-HCl pH8.0, 0,1 mM EDTA, 200 mM NaCl, 0,5% NP40, 10% glycerol, 20 mM N-ethylmaleimide (NEM), 1x Protease inhibitor cocktail tablets (Roche)). Plasma membrane proteins were enriched by harvesting cells in a non-denaturing lysis buffer (50 mM Tris-HCl, pH 7.4, 150 mM NaCl, 5 mM $MgCl_2$, 1 mM EDTA, Protease inhibitor cocktail tablets). Cells were disrupted by sonication (three times on ice), followed by centrifugation for 10 min at 1000 g (4°C) to remove the nuclear fraction (pellet). An ultracentrifugation (90 min at 100 000 g) was then performed on the supernatant, the pellet containing the membrane fraction was resuspended in lysis buffer. Equal amounts of protein were subjected to SDS-PAGE (4–15% Criterion TGX gradient protein gel, Bio-rad), transferred on nitrocellulose membrane (Trans-blot turbo, Bio-rad), and then immunobloted using the indicated primary antibodies. Anti-rabbit and anti-mouse antibodies conjugated with IR800 or IR680 dyes were used as secondary antibodies, and the infrared signal was integrated using an infrared imaging system (LI-COR Odyssey). The bands intensities were calculated using the software associated with the Odyssey system (Image studio).

## Immunoprecipitation

For immunoprecipitation of RhoGDIα, cells were lysed in buffer (50 mM Tris-HCl pH 8.0, 0,1 mM EDTA, 200 mM NaCl, 0,5% NP40, 10% glycerol, 20 mM NEM, 1x Protease inhibitor cocktail tablets (Roche)) and incubated for 2 hr at 4°C with anti-RhoGDIα antibody. Immune complexes were collected by incubation for 1 hr at 4°C with ProteinG/A sepharose beads (GE Healthcare) and washed three times in lysis buffer. Whole cell lysate (input) or cell lysates immunoprecipitated with anti-RhoGDIα were subjected to SDS-PAGE followed by immunoblotting with anti-RhoGDIα, anti-SUMO1 and anti-SUMO2/3.

## In vitro cleavage of SAE2 by calpain

Recombinant SAE2 (Novus biologicals) (10 μg) was digested with two different concentrations of calpain-1 (Merck Millipore; 0.2 and 2 U/mg) in reaction buffer (50 mM Tris-HCl, pH 7.5, 100 mM NaCl, 2 mM DTT, 1 mM EDTA, 3 mM $CaCl_2$) at 30°C for 20 min. The reaction was stopped by boiling samples for 5 min after the addition of an equal volume of 2X loading sample buffer (Bio-rad). Samples were then subjected to immunoblot analysis.

## SUMO-AMC assays

For non-denatured *Shigella* lysates, $4 \times 10^9$ bacteria were centrifuged and resuspended in 400 μL of lysis buffer (50 mM Tris, 150 mM NaCl, protease inhibitor cocktail (Roche), 0,5% Triton) and then sonicated. DTT was added to the lysate at a final concentration of 5 mM. Bacterial lysate was diluted to 1/10 within reaction buffer (50 mM Tris, 150 mM NaCl, 0,75 mg/mL BSA, 2 mM cysteine and 1 mg/mL Chaps). Recombinant GST fusion of the SENP2 catalytic domain was added in the reaction

buffer at a final concentration of 40 nM. SUMO1-AMC or SUMO-2-AMC (Boston Biochem) were added in the diluted sample at a final concentration of 100 nM, in a total volume of 200 μL. Liberation of AMC at room temperature during 60 min was monitored in a fluorimetric microplate reader (Infinite 200 pro, Tecan) with excitation at 380 nm and emission at 460 nm.

## Statistical methods

All experiments were performed at least three times. Statistical analyses were performed using two-tailed Student's t-test to calculate p-values. Statistical analyses on Pearson's correlation coefficients (Rr) were performed using a Fisher r-to-z transformation and then a two-tailed z-test on z values obtained.

## Ethic issues

Work on animals was conducted under animal study protocols #HA0042 approved by the Committeee of the Institut Pasteur for ethics in animal experimentation (CETEA) for its compliance with ethics rules (3Rs, cost-benefit balance), in application of the European Directive 2010/63/EU and of the derived French regulation.

## Acknowledgements

We acknowledge Chuanshu Huang and Mark R Philips for the generous gifts of RhoGDIα plasmids. We thank Peter A Greer for providing the *Capns1 WT* and *Capns1 KO* MEFs. We are grateful to Jacob S Seeler for providing unpublished auxin-inducible degron-UBC9 HT1080 cells and for intellectual input. We also thank Sandrine Etienne-Manneville for helpful discussions. This work was supported by grants from LNCC (Equipe labellisée), Odyssey-RE, INCa and ERC-AdG 'SUMOSTRESS'. PL was supported by LNCC. SF was supported by EEC 'RUBICON' and Sidaction.

## Additional information

### Funding

| Funder | Grant reference number | Author |
|---|---|---|
| Ligue Contre le Cancer | Post-doc fellowship | Pierre Lapaquette |
| Sidaction | | Sabrina Fritah |
| Institut Pasteur | | Philippe Sansonetti Anne Dejean |
| Institut National de la Santé et de la Recherche Médicale | | Philippe Sansonetti Anne Dejean |
| Institut National Du Cancer | | Anne Dejean |
| European Research Council | SUMOSTRESS | Anne Dejean |
| Odyssey RE | | Anne Dejean |
| Ligue Contre le Cancer | Labelled team | Anne Dejean |

The funders had no role in study design, data collection and interpretation, or the decision to submit the work for publication.

### Author contributions

Pierre Lapaquette, Conceptualization, Formal analysis, Investigation, Methodology, Writing—original draft, Writing—review and editing; Sabrina Fritah, Conceptualization, Formal analysis, Investigation, Methodology, Writing—review and editing; Nouara Lhocine, Giulia Nigro, Joëlle Mounier, Formal analysis, Methodology; Alexandra Andrieux, Methodology; Philippe Sansonetti, Conceptualization, Supervision, Funding acquisition, Investigation, Project administration, Writing—review and editing; Anne Dejean, Conceptualization, Formal analysis, Supervision, Funding acquisition, Investigation, Writing—original draft, Project administration, Writing—review and editing

**Author ORCIDs**

Pierre Lapaquette https://orcid.org/0000-0002-2680-351X

Anne Dejean http://orcid.org/0000-0003-4778-6840

**Ethics**

Animal experimentation: Animal experiments were performed accordingly to the guidelines of the Institut Pasteur's ethical committee for animal use in research (CETEA number 2013-0028).

**Decision letter and Author response**

Decision letter https://doi.org/10.7554/eLife.27444.026

Author response https://doi.org/10.7554/eLife.27444.027

## Additional files

**Supplementary files**

• Transparent reporting form

DOI: https://doi.org/10.7554/eLife.27444.022

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
