## [Decision Letter]

Thank you for submitting your article "*Shigella* entry unveils a calcium/calpain-dependent mechanism for inhibiting sumoylation" for consideration by *eLife*. Your article has been reviewed by three peer reviewers, and the evaluation has been overseen by a Reviewing Editor and Ivan Dikic as the Senior Editor. The following individuals involved in review of your submission have agreed to reveal their identity: Hubert Hilbi (Reviewer #3).

The reviewers have discussed the reviews with one another and the Reviewing Editor has drafted this decision to help you prepare a revised submission.

Summary:

The study by Lapaquette et al., represents an important advance in our understanding of mechanisms of host-pathogen communication and provides novel insight about how an invasive bacterial pathogen facilitates its uptake.

*Shigella flexneri*, the causative agent of bacillary dysentery hijacks the host ubiquitin-like system SUMOylation, Entry into epithelial cells is an obligatory event for *Shigella* virulence antagonized by SUMOylation. To dissect how *Shigella* interferes with the SUMO system they extend these findings to an in vivo model of infection (newborn mouse). Next, they explore a possible link between hyposumoylation and calcium activated proteases, calpains, which are known to be activated upon *Shigella* infection. The investigators show that upon infection of epithelial cells by *S. flexneri* (i) sumoylation is inhibited, and in turn, sumoylation inhibits bacterial invasion, (ii) the protease calpain and calcium levels regulate sumoylation, (iii) calpain cleaves the SUMO E1 enzyme SAE2, (iv) sumoylation of the Rho GTPase inhibitor RhoGDIα is blocked, and (v) sumoylated proteins and RhoGDIα are recruited to *S. flexneri* entry sites.

In summary, the study provides evidence that *S. flexneri*-triggered activation of calpain cleaves SAE2, reduces sumoylation of RhoGDIα and favors membrane accumulation of active Rho GTPases. Thus, in essence *S. flexneri* stimulates Rho GTPase activity, thereby facilitating cytoskeletal rearrangements and promoting its uptake by non-phagocytic host cells.

Essential revisions:

1) The authors showed in this and previous studies that (i) calpain cleaves SAE2, (ii) SAE2 knockdown promotes *S. flexneri* entry into host cells, and (iii) sumoylation of RhoGDIα regulates entry. To provide a direct link between SAE2 and RhoGDIα, the authors should assess whether depletion of SAE2 indeed reduces the sumoylation of RhoGDIα during infection with *S. flexneri*. Also, how relevant the observation of global hyposumoylation is in the context of *Shigella* infection as it seems that it is rather a direct response to changed Ca^2+^ levels that is a common consequence of many different stimuli?

2) It is unclear how the global desumoylation and specific RhoGDIα desumoylation, are linked. Clearly, the RhoGDIα desumoylation is more likely a consequence of increased desumolyation mediated perhaps by some bacterial effector than impaired sumoylation through inhibition of SAE2 by calpain-mediated degradation. Furthermore, the kinetics of the two events is very different with RhoGDIα desumoylation occurring early and global hyposumoylation later post-infection. The authors need to explain how RhoGDIα desumoylation and cytoskeleton rearrangements fit to the global desumoylation effect.

3) The dependence of RhoGDIα colocalization with *Shigella* on UBC9 is unconvincing. The authors should at least provide quantification of Figure 7. Also, see comment for Figure 6 similar experiment would strengthen the claim that RhoGDIα colocalization with *Shigella* depends on SUMO machinery and sumoylation.

4) The roles of *Shigella* factors, responsible for mediating the hyposumoylation are not adequately explored. VirA is conspicuous by its absence. A previous report (the one that links *Shigella* infection to calpain activation) reported that *Shigella*-induced calpain activation is VirA dependent. VirA has had multiple activities ascribed to it and this has resulted in a confusing body of literature. In their Discussion they state that they screened a collection of *Shigella* mutants and none of the mutants impaired hyposumoylation. They should show this data, and give some indication as to the nature of the "collection". Was it a collection of mutants that they had in house? Was it a library of transposon mutants? How was the experiment done? How many mutants? Was a virA mutant in the collection? The authors need to test if hyposumoylation is induced in a virA mutant and include the findings in this manuscript.

---

## [Author Response]

Essential revisions:1) The authors showed in this and previous studies that (i) calpain cleaves SAE2, (ii) SAE2 knockdown promotes S. flexneri entry into host cells, and (iii) sumoylation of RhoGDIα regulates entry. To provide a direct link between SAE2 and RhoGDIα, the authors should assess whether depletion of SAE2 indeed reduces the sumoylation of RhoGDIα during infection with S. flexneri.

The SAE1/SAE2 heterodimer is the unique E1 enzyme of the SUMO pathway so that loss of SAE1 or SAE2 invariably leads to the complete disappearance of all SUMO conjugates (as shown in Figure 5 of this study following SAE2 knock down). Moreover, we show that *Shigella* induces an identical loss in SUMO conjugates, including SUMO-RhoGDIα, via a calpain-dependent degradation of SAE2. So depleting SAE2 through siRNA or degrading SAE2 by *Shigella* infection results in the same outcome: global inhibition of sumoylation. We thus did not perform the experiment that would combine both approaches as each one, taken individually, leads to the complete loss in SUMO-modified proteins. However, to address the point raised by the reviewers of the direct link between the loss of SAE2 and the loss of SUMO-RhoGDIα, we depleted SAE2 through siRNA in Hela cells, thus mimicking *Shigella*-induced SAE2 disappearance, and then monitor the basal state of RhoGDIα sumoylation. Even upon forced sumoylation conditions (by overexpression of SUMO1), we observed that a decrease in SAE2 protein level results in a nearly complete loss of sumoylated RhoGDIα, without affecting the level of unmodified form (see Author response image 1).

**Author response image 1. respfig1:** Sumoylation of RhoGDIα is affected by SAE2 impairment. HeLa cells were treated with control siRNA (siScr) or siSAE2, and then co-transferred with expression vectors for GFP-RhoGDIα and SUMO1. Immunoblot analysis was performed on whole cell extracts using anti-GFP, -SAE2 and -Tubulin antibodies. Arrowhead indicates unmodified RhoGDlα and arrow indicates SUMO1- RhoGDIα./Author response image 1 title/legend>

Also, how relevant the observation of global hyposumoylation is in the context of Shigella infection as it seems that it is rather a direct response to changed Ca^2+^ levels that is a common consequence of many different stimuli?

This is a strong point and we appreciate the reviewers' comments here. Beyond the identification of RhoGDIα as an important SUMO substrate targeted by bacteria to enhance infectivity, the finding that global sumoylation can be regulated by modulation of intracellular calcium levels is indeed the second main message of this work. We believe the link between calcium and SUMO signaling discovered here provides a new conceptual framework for the regulatable nature of sumoylation, that goes well beyond the sole infection by *Shigella*. Indeed, as stressed by the reviewers, many physiological and pathological situations are associated with changes in cytosolic calcium levels. This result points out once again to the importance of pathogenic micro-organisms to unveil fundamental biochemical processes of the host cell. Moreover, *Shigella*-induced calcium response is highly specific in terms of kinetics, localization and intensity. Notably, the highly localized nature of the calcium response induced by *Shigella* at early time points of infection (Bonnet and Tran Van Nhieu, 2016) raises the intriguing possibility that sumoylation of a small subset of substrates (including sumoylation of RhoGDIα) can be affected locally during the early stages of infection, suggesting a novel paradigm for sumoylation.

With regards to the relevance of (Ca^2+^-induced) sumoylation inhibition in the context of *Shigella* infection, we had shown previously that sumoylation is an important negative regulator of *Shigella* infectivity (Fritah et al., 2014). Decreasing experimentally the global level of sumoylation in host cells favors *Shigella* spreading in vitro as well as bacterial invasion and intestinal mucosa inflammation in vivo. However, whether *Shigella* indeed targets the global SUMO pathway in host cells was unknown. Here we report that *Shigella* inhibits sumoylation and that this effect is mediated via a Ca^2+^/calpain-induced degradation of SAE2 which in turn facilitates *Shigella*-induced cytoskeletal rearrangements and bacterial entry.

2) It is unclear how the global desumoylation and specific RhoGDIα desumoylation, are linked. Clearly, the RhoGDIα desumoylation is more likely a consequence of increased desumolyation mediated perhaps by some bacterial effector than impaired sumoylation through inhibition of SAE2 by calpain-mediated degradation. Furthermore, the kinetics of the two events is very different with RhoGDIα desumoylation occurring early and global hyposumoylation later post-infection. The authors need to explain how RhoGDIα desumoylation and cytoskeleton rearrangements fit to the global desumoylation effect.

We agree with the reviewers that the link between the global inhibition of sumoylation, the specific loss of SUMO-RhoGDIα and the calpain-induced degradation of SAE2 is an important issue. One may distinguish here the kinetics of loss of SUMO-RhoGDIα, global SUMO conjugates and SAE2.

The loss of SUMO-RhoGDIα coincides with the first visible reduction in the amount of total sumoylated proteins after 30 min (Figure 1 and Figure 1—figure supplement 1). Quantification showing a slight, yet consistant ~25% reduction in the level of SUMO1 conjugates and concomitant accumulation of free SUMO1 after 30 min has been added in the new Figure 1—figure supplement 1 and in the Results section. The strong decrease in the total amount of SUMO2-conjugates after 30 min was readily shown in the original Figure 1—figure supplement 1. The early global inhibition of sumoylation and the specific loss of SUMO-RhoGDIα are thus directly linked. The rapidity with which the sumoylated form of a given protein is lost is highly specific for the substrate and relies on its half-life and the dynamics of its proper sumoylation/desumoylation equilibrium which also depends on its intracellular localization. The impairment of sumoylation will result in a gradual loss of SUMO conjugates starting with the most sensitive substrates, such as RhoGDIα, and will ultimately affect the most heavily and stably modified ones like RanGAP1. To remove the original ambiguous formulation and make this clearer in the revision, we now mention: “Whereas the levels of unmodified RhoGDIα remained unaffected, we observed the complete disappearance of SUMO-RhoGDIα as quickly as 30 min post-infection (Figure 8). The reduction in the global amount of SUMO1 conjugates was only moderate in these conditions (Figure 1 and Figure 1—figure supplement 1), indicating that RhoGDIα is highly sensitive to sumoylation inhibition” (subsection “Endogenous RhoGDIα and SUMO proteins are recruited at bacteria entry sites and *Shigella* rapidly impairs RhoGDIαsumoylation”).

We agree with the reviewers that there is an apparent discordance between the loss of SAE2, which is not yet visible 1 h post-infection (Figure 4), and the reduction in global sumoylation, that begins 30 min after infection. This point has been now clearly stated in the Results section and extensively discussed (Discussion section). Because of this readily noticeable time lag for SAE2 degradation, we feel that earlier time points between 0 and 1 h would not be really informative. The mechanisms responsible for the early reduction in SUMO conjugates triggered by *Shigella* infection remain to be elucidated. As pointed by the reviewers, *Shigella* may have developed additional strategies to impair sumoylation early during infection, such as activation of desumoylases. However, a direct involvement of increased SENP activities seems unlikely as the loss in SUMO-modified proteins is, like SAE2 loss, strictly dependent on calpains (Figure 2), and no link has been so far established between calpain and SENP activities. Moreover, we have now measured the global desumoylase activity in *Shigella* M90T non-denatured lysates. Using fluorogenic 7-amino-4-methylcoumarin SUMO substrates (SUMO1-AMC and SUMO2-AMC), we failed to reveal any SUMO1 or SUMO2 protease activities in M90T extracts when compared to recombinant SENP2 used here as a positive control. The data are shown in the new Figure 4—figure supplement 1.

Since the loss of SAE2 and SUMO conjugates are both calpain-dependent, we favor the hypothesis that the rapid loss of SUMO-RhoGDIα results from the first and localized wave of Ca^2+^ response induced by *Shigella* in the vicinity of the entry sites. This response would lead to a local calpain-induced degradation of a small pull of SAE2, hardly detectable by Western blot, and subsequent loss of highly sensitive SUMO conjugates including SUMO-RhoGDIα. A more definitive clarification of this issue must, however, await the development of probes permitting the visualization of localized sumoylation dynamics. We have modified and expanded the corresponding section in the Discussion section to comment more in depth on this point. Since we cannot strictly rule out that non-SAE2-dependent mechanisms are involved in the early hyposumoylation triggered by *Shigella*, we have also added the word 'mainly' in the sentence “this effect is mainly mediated by a calcium/calpain-induced cleavage of the SUMO E1 enzyme SAE2” in the Abstract.

3) The dependence of RhoGDIα colocalization with Shigella on UBC9 is unconvincing. The authors should at least provide quantification of Figure 7. Also, see comment for Figure 6 similar experiment would strengthen the claim that RhoGDIα colocalization with Shigella depends on SUMO machinery and sumoylation.

We thank the reviewers for this comment. As requested, quantification has been performed by using the colocalization studio plugin of Icy software (Lagache et al., 2015), enabling the calculation of Pearson's correlation coefficients (Rr) between fluorescent signals from SUMO1, SUMO2/3 or RhoGDIα immunostaining and signal from actin foci at *Shigella* entry site. We have included these data as new figure panels 7B, D and F and in the corresponding Results section. Analysis in Ubc9 WT MEFs reveals a moderate positive correlation between SUMO1 (Rr=0.41, p=0.008), SUMO2/3 (Rr=0.36, p=0.025), RhoGDIα (Rr=0.47, p=0.000017) and actin at bacteria entry site (at least 40 entry foci analysed per condition), confirming an enrichment in SUMO1, SUMO2/3 and RhoGDIα at *Shigella* entry foci in Ubc9 WT-infected MEFs. These positive correlations are weaker and lose significance for SUMO1 (Rr=0.226, p=0.073) and SUMO2/3 (Rr=0.290, p=0.082) in Ubc9 KO cells, indicating a decrease in conjugated SUMO signal at *Shigella* entry site in cells depleted in sumoylation. The positive correlation between RhoGDIα immunostaining and *Shigella* entry sites remains statistically unchanged upon loss of sumoylation (Rr= 0.487, p=0.0006), indicating that, in contrast to what we suggested initially, RhoGDIα colocalization with *Shigella* entry foci does not depend on an active SUMO machinery. We have corrected these data in the Results section.

4) The roles of Shigella factors, responsible for mediating the hyposumoylation are not adequately explored. VirA is conspicuous by its absence. A previous report (the one that links Shigella infection to calpain activation) reported that Shigella-induced calpain activation is VirA dependent. VirA has had multiple activities ascribed to it and this has resulted in a confusing body of literature. In their Discussion they state that they screened a collection of Shigella mutants and none of the mutants impaired hyposumoylation. They should show this data, and give some indication as to the nature of the "collection". Was it a collection of mutants that they had in house? Was it a library of transposon mutants? How was the experiment done? How many mutants? Was a virA mutant in the collection? The authors need to test if hyposumoylation is induced in a virA mutant and include the findings in this manuscript.

To identify putative *Shigella* factors involved in sumoylation inhibition, we have now used a wider in-house panel of mutant strains: *mxiD, mxiE, ospG, ipaB, ipaC, ipaC*/pC35, *ipaC*/pC57, *ipgD* and *virA*. Data are presented in the new Figure 1—figure supplement 2. Among these mutants, the strain mutated for the transcriptional activator MxiE is still able to induce a loss in SUMO conjugates, despite the fact that this mutant is unable to express and secrete numerous *Shigella* effectors and some IpaH proteins encoded by chromosomal genes (Bongrand et al., 2012; Kane et al., 2002; Mavris et al., 2002). Similar observations were made for the strains defective for the IpgD and OspG effectors. In contrast, mutants for the genes encoding the translocator proteins IpaB and IpaC are no longer able to trigger loss of SUMO conjugates, a feature shared with the *mxiD* mutant. These findings indicate the involvement of T3SS-related processes in *Shigella*-induced sumoylation inhibition. Interestingly, IpaB and IpaC proteins act as translocator-forming proteins that are able to target the host plasma membrane. As described in the Discussion section, three pore forming toxins produced by Gram positive bacteria have already been reported to induce a massive loss in SUMO conjugates (Ribet et al., 2010), thus suggesting a similar mechanism for the Gram negative *Shigella*. The use of two insertion mutants of IpaC, *ipaC*/pC57 and *ipaC*/pC351 (Mounier et al., 2009), further revealed that the stress at the plasma membrane triggered by this translocator protein, appears to be sufficient to induce the loss in SUMO conjugates. Indeed, the *ipaC*/pC57 mutant, that is defective for invasion but still able to induce actin foci formation at the host plasma membrane, induces hyposumoylation, whereas the *ipaC*/pC351 mutant, defective for actin foci formation, is unable to do so. Calpain proteases have been amply described to participate in the healing of damaged plasma membranes by allowing local remodeling of the cortical cytoskeleton (Godell et al., 1997; Mellgren et al., 2007; Mellgren et al., 2009). One may thus hypothesize that, by triggering pore formation and subsequent plasma membrane stress, *Shigella* infection induces a calpain-dependent loss of sumoylation. We have included these data as new Figure 1—figure supplement 2 and in the Results and Discussion sections.

With regards to the effect of VirA pointed out by the reviewers, Bergounioux et al., (2012) have reported that the virA defective strain is still able to induce calpain activation, albeit less rapidly than the wild-type strain. In our hands, the *mxiE* mutant, that lacks expression of VirA, or the *virA* mutant itself were as potent as the wild-type strain to induce calpain activation, SAE2 degradation and the subsequent loss in SUMO1 and SUMO2/3 conjugates 2h post-infection (see Author response image 2).

**Author response image 2. respfig2:** The virA defective mutant induces calpain activation and loss of SUMO conjugates and SAE2. SUMO1 and SUMO2 patterns in HeLa cells, uninfected or infected for 120min with the wild-type *Shigella* strain M90T or mutants *mxiD* or *virA*. Immunoblot analysis was performed on whole-cell lysates using antibodies specific for SUMO1, SUMO2/3, Capns1, SAE2 and Tubulin. Arrows indicate autolytic fragments of Capns1. Tubulin was used as a loading control.